# Graph Fourier Neural ODEs: Modeling Spatial-temporal Multi-scales in Molecular Dynamics

**Fang Sun**                                                            *fts@cs.ucla.edu*
*University of California, Los Angeles*

**Zijie Huang**                                                  *zijiehuang@cs.ucla.edu*
*University of California, Los Angeles*

**Haixin Wang**                                                        *whx@cs.ucla.edu*
*University of California, Los Angeles*

**Huacong Tang**                                                     *hctang@ucla.edu*
*University of California, Los Angeles*

**Xiao Luo**                                                        *xiaoluo@cs.ucla.edu*
*University of California, Los Angeles*

**Wei Wang**                                                       *weiwang@cs.ucla.edu*
*University of California, Los Angeles*

**Yizhou Sun**                                                      *yzsun@cs.ucla.edu*
*University of California, Los Angeles*

**Reviewed on OpenReview:** *https://openreview.net/forum?id=XK7cIdj6Fz*

## Abstract

Accurately predicting long-horizon molecular dynamics (MD) trajectories remains a significant challenge, as existing deep learning methods often struggle to retain fidelity over extended simulations. We hypothesize that one key factor limiting accuracy is the difficulty of capturing interactions that span distinct spatial and temporal scales—ranging from high-frequency local vibrations to low-frequency global conformational changes. To address these limitations, we propose **Graph Fourier Neural ODEs (GF-NODE)**, integrating a graph Fourier transform for spatial frequency decomposition with a Neural ODE framework for continuous-time evolution. Specifically, GF-NODE first decomposes molecular configurations into multiple spatial frequency modes using the graph Laplacian, then evolves the frequency components in time via a learnable Neural ODE module that captures both local and global dynamics, and finally reconstructs the updated molecular geometry through an inverse graph Fourier transform. By explicitly modeling high- and low-frequency phenomena in this unified pipeline, GF-NODE more effectively captures long-range correlations and local fluctuations alike. We provide theoretical insight through heat equation analysis on a simplified diffusion model, demonstrating how graph Laplacian eigenvalues can determine temporal dynamics scales, and crucially validate this correspondence through comprehensive empirical analysis on real molecular dynamics trajectories showing quantitative spatial-temporal correlations across diverse molecular systems. Experimental results on challenging MD benchmarks, including MD17 and alanine dipeptide, demonstrate that GF-NODE achieves state-of-the-art accuracy while preserving essential geometrical features over extended simulations. These findings highlight the promise of bridging spectral decomposition with continuous-time modeling to improve the robustness and predictive power of MD simulations. Our implementation is publicly available at `https://github.com/FrancoTSolis/GF-NODE-code`.

# 1 Introduction

Molecular dynamics (MD) simulations are indispensable tools for investigating the behavior of molecular systems at the atomic level, offering profound insights into physical (Bear & Blaisten-Barojas, 1998), chemical (Wang et al., 2011), and biological (Salo-Ahen et al., 2020) processes. These simulations must capture interactions occurring across a wide range of spatial and temporal scales—from localized bond vibrations to long-range non-bonded interactions—posing significant computational challenges. Accurately modeling these multiscale interactions is crucial for uncovering the mechanisms underlying complex molecular phenomena but remains prohibitively expensive for large systems and long trajectories (Vakis et al., 2018).

In recent years, Graph Neural ODEs (Zang & Wang, 2020; Huang et al., 2023a) have gained traction for modeling continuous-time dynamics in multi-agent systems, including MD. By learning an Ordinary Differential Equation (ODE) function via Graph Neural Networks and solving it numerically, these methods allow flexible sampling at arbitrary time points. Such a continuous-time formulation is well-suited for capturing multiple temporal scales inherent in molecular simulations. However, significant challenges persist in accounting for the rich spatial multiscale effects that span localized bond vibrations to extended nonbonded interactions. On another front, Fourier Neural Operators (FNOs) (Li et al., 2020) have demonstrated success in learning operators by decomposing signals into different frequency modes, thereby capturing various spatial scales effectively. Yet, they are not tailored for graph-structured molecular data or continuous-time temporal evolution, limiting their direct applicability to general MD simulations.

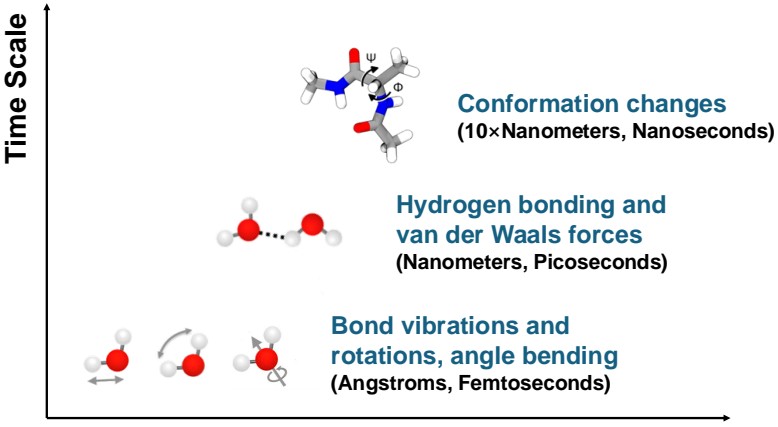

Figure 1: Illustration of spatial and temporal multiscale interactions in molecular dynamics. The x-axis indicates spatial scales ranging from local bond vibrations to global conformational changes. The y-axis represents timescales ranging from femtoseconds (bond oscillations) to nanoseconds (conformational rearrangements). Local interactions, such as bond vibrations and angle bending, occur over short spatial scales (angstroms) and fast timescales (femtoseconds). Non-local interactions, including hydrogen bonding and van der Waals forces, span larger spatial scales (nanometers) and intermediate timescales (picoseconds). Conformational changes involve the longest spatial and temporal scales, potentially up to nanoseconds or longer.

Despite the promise of Graph Neural ODEs for handling multiple temporal scales via continuous time modeling, they alone are inadequate for fully capturing the complex spatial frequency components of molecular systems. Conversely, while approaches based on Fourier transforms can model multiple spatial scales, they do not naturally handle the intricacies of molecular graphs or continuous-time dynamics. To address these limitations, we introduce *Graph Fourier Neural ODEs (GF-NODE)*. As demonstrated in Figure 2, our framework explicitly integrates a graph Fourier transform—for decomposing and encoding spatial multiscale interactions—with a Neural ODE framework for continuous-time modeling of each spatial frequency. By leveraging an inverse graph Fourier transform at the end of the pipeline, GF-NODE reconstructs the molecular state in physical space, thereby enabling a unified approach to spatial and temporal multiscale simulation.

We conduct extensive experiments on benchmark molecular dynamics datasets, including MD17 and alanine dipeptide. Empirical results show that GF-NODE achieves state-of-the-art accuracy in predicting molecular trajectories over long-horizon simulations, preserves essential geometric properties such as bond lengths and angles, and demonstrates stable performance over temporal super-resolution tasks. These findings underscore the importance of explicitly decomposing molecular configurations into spatial frequency modes and evolving

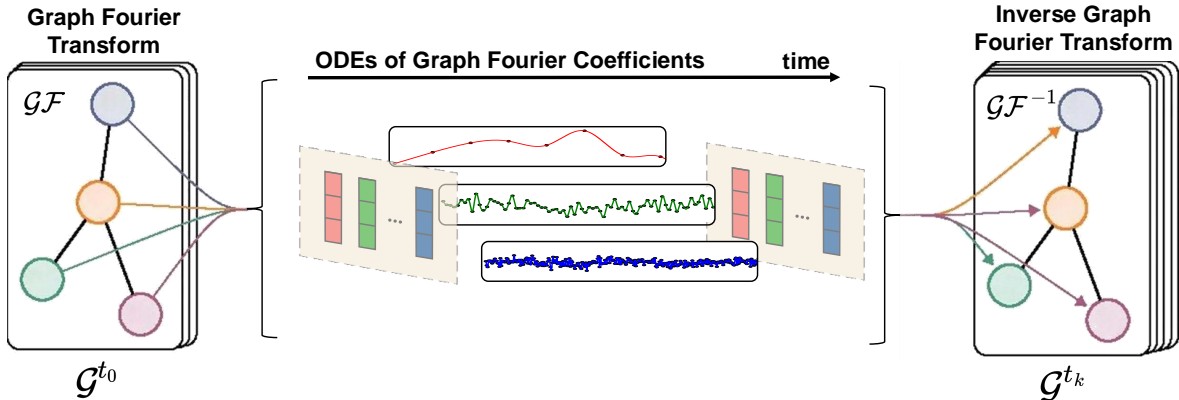

Figure 2: Overview of the Graph Fourier Neural ODE framework. The molecular graph $\mathcal{G}^{t_0}$ is first transformed into the spectral domain using a Graph Fourier Transform ($\mathcal{GF}$), decomposing the spatial structure into frequency components. Neural ODEs are then applied to evolve the Fourier coefficients over time. The evolved coefficients are finally transformed back into the physical domain using an inverse Graph Fourier Transform ($\mathcal{GF}^{-1}$), reconstructing the molecular graph at future time $t_k$.

them continuously in time. Our analysis suggests that this multiscale perspective is instrumental for capturing both rapid local fluctuations and slow global conformational changes.

Our contribution can be summarized as follows.

*(a) New perespective.* We provide a new perspective on spatial-temporal multiscale modeling for molecular dynamics by jointly capturing spatial and temporal interactions within a single framework.

*(b) Novel architecture.* Building on this perspective, we introduce the *Graph Fourier Neural ODEs (GF-NODE)* architecture, which combines a graph Fourier transform for decomposing molecular interactions into distinct spatial frequencies with a Neural ODE formulation to evolve these frequencies continuously in time.

*(c) Theoretical foundations.* We establish theoretical insight into the correspondence between spatial frequency modes and temporal dynamics scales through heat equation analysis on a simplified diffusion model, showing that graph Laplacian eigenvalues can intrinsically determine temporal evolution rates. Importantly, we provide comprehensive empirical validation through our Theoretical Framework for Joint Spatial-Temporal Analysis, demonstrating quantitative spatial-temporal correlations on real molecular dynamics trajectories across diverse systems.

*(d) Comprehensive evaluation.* We conduct extensive experiments across multiple benchmarks, including the Revised MD17 dataset, larger molecular systems (20-326 heavy atoms), and extended temporal horizons up to 15,000 MD steps, consistently achieving state-of-the-art performance with detailed structural analysis confirming accurate preservation of molecular correlations.

## 2 Related Work

We review relevant works on multi-scale modeling in molecular dynamics, focusing on neural operator models and graph neural ODEs.

### 2.1 Classical Molecular Simulation Methods

Traditional molecular simulation methods, including force-field based MD (e.g., AMBER (Cornell et al., 1995; Brooks et al., 2009), GAMD (Li et al., 2022b)) and *ab initio* techniques such as Car–Parrinello MD (Hutter, 2012), have been foundational in exploring molecular behavior. However, these methods face significant

limitations: force-field simulations require extremely small time steps to accurately resolve high-frequency bond vibrations, which hampers long-term stability and computational efficiency; *ab initio* MD, though offering first-principles accuracy, is computationally prohibitive for large systems and long trajectories; and while coarse-grained models (e.g., MARTINI (Souza et al., 2021)) enable more efficient multiscale simulations, they often compromise on molecular detail and accuracy, particularly in reproducing local interactions and maintaining seamless force consistency at multiscale interfaces.

## 2.2 Neural Operator Models

Neural operator models (Kovachki et al., 2023) have emerged as powerful tools for learning mappings between infinite-dimensional function spaces, demonstrating success in modeling complex dynamical systems. Among these, *Fourier Neural Operators (FNOs)* (Li et al., 2020; 2022a; Liu & Jafarzadeh, 2023; Kovachki et al., 2021; Koshizuka et al.) are particularly notable for handling spatial multi-scale interactions in partial differential equation (PDE) data by learning representations in the Fourier domain. However, while FNOs efficiently capture spatial hierarchies, they do not inherently model temporal dynamics, making them suboptimal for time-evolving molecular systems.

In contrast, recent operator-based methods such as the *Implicit Transfer Operator (ITO)* (Schreiner et al., 2024), *Timewarp* (Klein et al., 2024), and *Equivariant Graph Neural Operator (EGNO)* (Xu et al., 2024) focus on temporal multi-scale modeling in molecular dynamics. ITO and Timewarp introduce coarse-graining and adaptive time-stepping mechanisms to accelerate long-horizon simulations. EGNO employs neural operators with SE(3) equivariance to capture rotational and translational symmetries, yet it primarily addresses temporal evolution without explicitly handling spatial multi-scale effects. While these models successfully extend the applicability of neural operators to molecular simulations, none jointly addresses both spatial and temporal multi-scales in molecular dynamics.

## 2.3 Graph Neural ODE Models

Graph Neural ODEs combine Graph Neural Networks with Neural ODE frameworks (Chen et al., 2018; Kidger, 2022; Goyal & Benner, 2023; Holt et al., 2022; Luo et al., 2023; Luo et al.) to model continuous-time dynamics on graph-structured data (Zang & Wang, 2020; Kim et al., 2021; Huang et al., 2020; 2021; 2023a;b). These methods excel in capturing temporal multi-scale behavior by allowing flexible time integration, making them well-suited for systems with varying temporal resolutions. However, they primarily focus on modeling temporal dependencies, with little emphasis on explicitly handling spatial multi-scales in molecular dynamics.

A key limitation of existing Graph Neural ODE models is their reliance on local message passing, which inherently constrains their ability to capture long-range spatial dependencies within molecular systems. As a result, they may fail to adequately represent the interplay between localized, high-frequency interactions (e.g., bond vibrations) and global, low-frequency effects (e.g., large-scale conformational changes). Unlike spectral-based approaches that can decompose spatial hierarchies, standard Graph Neural ODEs lack a mechanism to explicitly encode spatial multi-scale structures, limiting their effectiveness in modeling complex molecular dynamics.

## 3 The Proposed Approach

We propose **GF-NODE**, a framework specifically designed to address the limitations of existing methods in capturing both spatial and temporal multiscale dynamics in molecular systems. As discussed in Sections 1 and 2, current approaches either handle spatial scales using Fourier-based methods or focus on temporal scales using Graph Neural ODEs, but they do not jointly model these scales in a unified framework. GF-NODE directly addresses this gap by integrating the *Graph Fourier Transform (GFT)* and *Neural Ordinary Differential Equations (Neural ODEs)* to simultaneously decompose spatial interactions and model their continuous-time evolution.

Specifically, as illustrated in Figure 3, GF-NODE first applies a Graph Fourier Transform to decompose the molecular graph into different frequency components, effectively separating localized, high-frequency

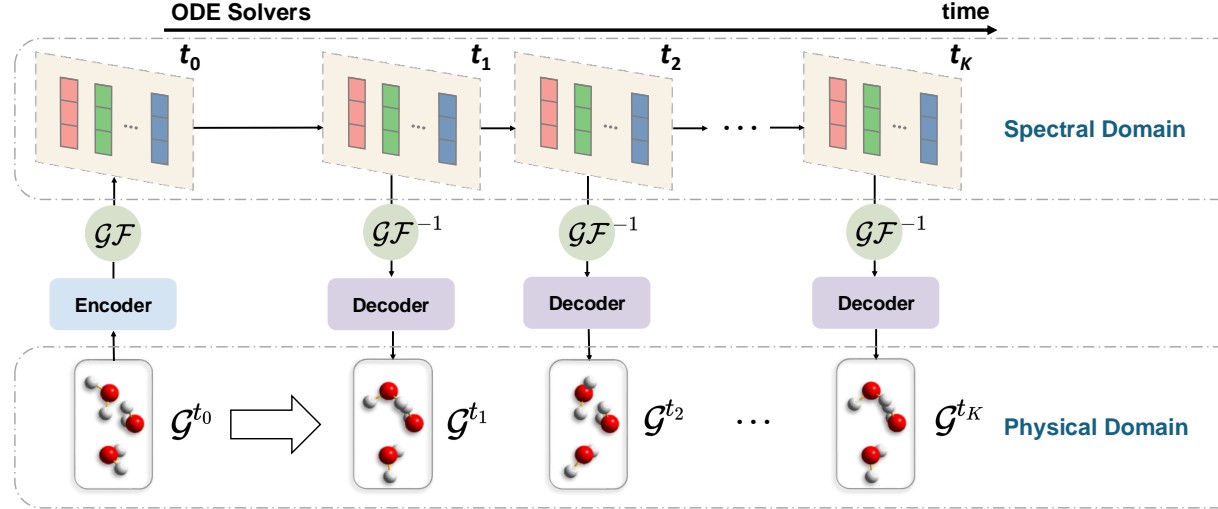

Figure 3: An overview of the proposed GF-NODE architecture. The model first encodes the initial molecular graph $\mathcal{G}^{t_0}$ from the physical domain into the spectral domain via a graph Fourier transform ($\mathcal{GF}$). Neural ODE solvers then propagate the dynamics of the Fourier coefficients continuously across time points $t_0, t_1, \ldots, t_K$. The transformed coefficients are subsequently decoded back to the physical domain using an inverse graph Fourier transform ($\mathcal{GF}^{-1}$), reconstructing the molecular graphs $\mathcal{G}^{t_1}, \mathcal{G}^{t_2}, \ldots, \mathcal{G}^{t_K}$ at future time points. This design enables efficient modeling of spatial and temporal multiscale dynamics in molecular systems.

interactions from global, low-frequency patterns. This spectral representation allows the model to process molecular structures in a frequency-adaptive manner, capturing both fine-grained local interactions and large-scale conformational changes. The decomposed spectral coefficients are then evolved continuously over time using Neural ODEs, ensuring flexible and adaptive modeling of multiscale temporal dynamics. Finally, the inverse Graph Fourier Transform reconstructs the molecular graph in the physical domain, preserving both local and global structural information over long-horizon molecular simulations.

The theoretical motivation for this design stems from heat equation analysis on simplified diffusion models, which suggests that spatial frequency modes (characterized by graph Laplacian eigenvalues) naturally correspond to different temporal evolution scales. Importantly, we provide comprehensive empirical validation of this correspondence through detailed spatial-temporal correlation analysis on real molecular dynamics trajectories, confirming that the spectral decomposition effectively separates distinct temporal dynamics scales in practice.

This design enables GF-NODE to overcome key challenges in molecular dynamics modeling: (1) explicitly encoding spatial multi-scale interactions via spectral decomposition, (2) leveraging continuous-time evolution to capture complex temporal dependencies, and (3) integrating these spatial and temporal scales into a single end-to-end framework that maintains SE(3) equivariance (formal proof provided in Appendix D). Crucially, our approach is grounded in rigorous theoretical foundations: graph Laplacian eigenvalues intrinsically determine the characteristic time-scales of dynamics through heat equation analysis, establishing that low-frequency spatial modes naturally correspond to slow temporal evolution while high-frequency modes correspond to rapid dynamics (detailed analysis in Section 4.4 and Appendix). Below, we detail the core components and operations of GF-NODE.

### 3.1 Notation and problem setup

Let $\mathcal{G} = (\mathcal{V}, \mathcal{E})$ represent a molecular graph, where $N$ is the number of nodes $i \in \{1, \ldots, N\}$ (atoms), and $\mathcal{E}$ is the set of edges representing chemical bonds or interactions. Edges are identified by checking whether the distance between atoms falls below a threshold. Each node $i$ has:

- Invariant (scalar) features $\mathbf{h}_i \in \mathbb{R}^F$, which include the velocity magnitude $\|\mathbf{v}_i\|$ and a normalized version of the atomic number $Z_i$;
- Vector features $\mathbf{z}_i \in \mathbb{R}^{m \times 3}$, which contain position $\mathbf{x}_i$ and velocity $\mathbf{v}_i$ (i.e. $\mathbf{z}_i = (\mathbf{x}_i, \mathbf{v}_i)$ when $m = 2$).

At time $t_0$, the molecular system's state is given as $\{\mathbf{h}, \mathbf{z}\}$, and we aim to predict the future configuration $\mathcal{G}^{(t_k)}$ for any $t_k > t_0$.

## 3.2 Graph Neural Network encoder

We use a Graph Neural Network (GNN) to encode scalar features $\mathbf{h}_i \in \mathbb{R}^F$ and vector features $\mathbf{z}_i \in \mathbb{R}^{m \times 3}$ for each node $i$. The initial scalar feature $\mathbf{h}_i^{(0)}$ for each atom $i$ is formed by concatenating $\|\mathbf{v}_i\|$ and $\frac{Z_i}{Z_{\max}}$, where $Z_i$ is the atomic number of atom $i$, and $Z_{\max}$ is a reference maximum. This concatenated vector is then mapped by a linear embedding layer, producing the hidden dimension used by the GNN. The vector feature $\mathbf{z}_i^{(0)}$ contains $\mathbf{x}_i$ and $\mathbf{v}_i$. Each **GNN** layer performs message passing, where node $i$'s features are updated based on its neighbors $\mathcal{N}(i)$ :

$$\mathbf{m}_{ij}^{(l)} = \phi_e\left(\mathbf{h}_i^{(l)}, \mathbf{h}_j^{(l)}, \mathbf{r}_{ij}^{(l)}\right), \tag{1}$$

$$\mathbf{h}_i^{(l+1)} = \phi_h\left(\mathbf{h}_i^{(l)}, \sum_{j \in \mathcal{N}(i)} \mathbf{m}_{ij}^{(l)}\right), \tag{2}$$

$$\mathbf{x}_i^{(l+1)} = \mathbf{x}_i^{(l)} + \frac{1}{|\mathcal{N}(i)|} \sum_{j \in \mathcal{N}(i)} \psi\left(\mathbf{m}_{ij}^{(l)}\right), \tag{3}$$

where $\mathbf{r}_{ij}^{(l)} = \mathbf{x}_i^{(l)} - \mathbf{x}_j^{(l)}$, and $\phi_e, \phi_h$, and $\psi$ are neural networks. Similarly, we also update the velocity $\mathbf{v}_i^{(l)}$ if present. After $L$ layers, the GNN produces the encoded features $\mathbf{h}_i^{(L)}$ and $\mathbf{z}_i^{(L)}$, capturing both local and global molecular information. These features are then passed to the Graph Fourier Transform (GFT) for spectral decomposition.

## 3.3 Graph Fourier Transform

Molecular dynamics exhibit behavior on multiple spatial scales: large-scale "global" deformations can be viewed as low-frequency modes, whereas fast "local" vibrations correspond to high frequency modes. In classical signal processing, Fourier analysis decomposes signals into sinusoids of different frequencies. Similarly, on graphs, we can decompose node-based signals into eigenmodes of a suitable operator (often the graph Laplacian). By retaining or emphasizing certain frequency bands, one can explicitly model global vs. local patterns.

**Graph Signal as Combination of Laplacian Eigenvectors.** A scalar function $f : \mathcal{V} \to \mathbb{R}$ on the nodes can be seen as a vector $\mathbf{f} \in \mathbb{R}^N$. The graph Laplacian $\mathbf{L} = \mathbf{D} - \mathbf{A}$ (or a symmetrized variant) admits an eigen-decomposition:

$$\mathbf{L} = \mathbf{U}\mathbf{\Lambda}\mathbf{U}^\top, \tag{4}$$

where $\mathbf{U} = [\mathbf{u}_0, \mathbf{u}_1, \ldots, \mathbf{u}_{N-1}]$ is an orthonormal matrix of eigenvectors, i.e. $\mathbf{u}_k \in \mathbb{R}^N$, and $\mathbf{\Lambda} = \mathrm{diag}\left(\lambda_0, \lambda_1, \ldots, \lambda_{N-1}\right)$ contains ascending eigenvalues $0 \le \lambda_0 \le \lambda_1 \le \cdots \le \lambda_{N-1}$.

In graph signal processing, the eigenvector $\mathbf{u}_k$ is viewed as the "$k$-th frequency basis." Smaller eigenvalues ($\lambda_0, \lambda_1, \ldots$) correspond to low-frequency (more global, smooth) variations on the graph, while larger eigenvalues correspond to high-frequency (local, rapidly changing) modes. Hence, any signal $\mathbf{f} \in \mathbb{R}^N$ can be written as a linear combination:

$$\mathbf{f} = \sum_{k=0}^{N-1} \alpha_k \mathbf{u}_k, \quad \text{where } \alpha_k = \mathbf{u}_k^\top \mathbf{f}. \tag{5}$$

This collection of coefficients $\{\alpha_k\}$ is the Graph Fourier Transform (GFT) of $\mathbf{f}$. We **now denote** $\alpha_k$ as $\tilde{f}_k$.

**Truncation at $M$ Bases.**   For large $N$, we often use only the first $M$ eigenvectors $\{\mathbf{u}_0, \ldots, \mathbf{u}_{M-1}\}$ to approximate $\mathbf{f}$. This yields a band-limited or multiscale representation:

$$\mathbf{f} \approx \sum_{k=0}^{M-1} \alpha_k \mathbf{u}_k, \tag{6}$$

where $M \leq N$. Sorting $\lambda_k$ in ascending order ensures the lowest-frequency modes (i.e., global scales) appear first, and higher-frequency (local scales) modes appear last. By choosing $M$ appropriately, we focus on the most critical modes for modeling global vs. local behavior.

**Applying the GFT to Scalar vs. Vector Features.**   For *scalar* features $\mathbf{H} \in \mathbb{R}^{N \times F}$, where $F$ is the latent feature size, we substitute $\tilde{f}_k$ for $\tilde{\mathbf{H}}_k$ in the equation $\tilde{f}_k = \mathbf{u}_k^\top f$ (from Equation 5) to get:

$$\tilde{\mathbf{H}} = \left[\tilde{\mathbf{H}}_k\right]_{k=0}^{M-1} = \mathbf{U}_{(:,0:M)}^\top \mathbf{H}, \tag{7}$$

where $\mathbf{U}_{(:,0:M)}$ denotes the first $M$ eigenvectors. For *vector* features $\mathbf{Z} \in \mathbb{R}^{N \times m \times 3}$, where $m$ is the feature size of the vector feature in the 3D space, we first remove the mean to ensure translational invariance (since the 0-th eigenvector $\mathbf{u}_0$ corresponds to the constant mode):

$$\mathbf{Z}_c = \mathbf{Z} - \overline{\mathbf{Z}}. \tag{8}$$

where $\overline{\mathbf{Z}}$ is the global mean over all nodes. We then apply the same truncated basis $\mathbf{U}_{(:,0:M)}^\top$ to each coordinate dimension, substituting $\tilde{f}_k$ for $\tilde{\mathbf{Z}}_k$ in the equation $\tilde{f}_k = \mathbf{u}_k^\top f$ (from Equation 5) to get:

$$\tilde{\mathbf{Z}} = \mathbf{U}_{(:,0:M)}^\top \mathbf{Z}_c \in \mathbb{R}^{M \times m \times 3}. \tag{9}$$

Indices closer to $k = 0$ indicate more global motions, while larger $k$ (up to $M - 1$) captures more local, high-frequency fluctuations. In practice, $M$ can be a hyperparameter that determines how many eigenmodes we keep, balancing efficiency (fewer modes to evolve) and accuracy (how many scales are captured).

**Theoretical Analysis.**   The Graph Fourier Transform (GFT) decomposes molecular dynamics into modes corresponding to global and local spatial patterns, as determined by the eigenvalues and eigenvectors of the graph Laplacian. As demonstrated in Proposition 3.1, low-frequency modes capture smooth, global deformations, while high-frequency modes represent localized structural variations. This decomposition enables efficient modeling of multiscale spatial dynamics.

**Proposition 3.1** (Global vs. Local Spatial Scales). *Let $\mathbf{u}_k$ be the $k$-th eigenvector of $\mathbf{L}$ with eigenvalue $\lambda_k$. Suppose $\mathbf{x}$ encodes atomic coordinates or their latent features. Then:*

1. ***If $\lambda_k$ is small**, the corresponding mode $\mathbf{u}_k$ represents slowly varying (global) deformations across the molecule.*
2. ***If $\lambda_k$ is large**, the corresponding mode $\mathbf{u}_k$ represents rapidly changing (local) structural variations.*

Regarding the number of modes $M$ we need to use, Theorem 3.2 guarantees that we can use the first few modes to represent the entire system accurately:

**Theorem 3.2** (Spectral Truncation Error). *Truncating the spectral representation to the first $M$ modes, $\mathbf{x}_{(M)} = \mathbf{U}_{(:,0:M)} \mathbf{U}_{(:,0:M)}^\top \mathbf{x}$, yields an $\ell^2$-norm error:*

$$\left\| \mathbf{x} - \mathbf{x}_{(M)} \right\|_2^2 = \sum_{k=M}^{N-1} \left| \mathbf{u}_k^\top \mathbf{x} \right|^2. \tag{10}$$

*For $\alpha$-bandlimited signals, choosing $M$ such that $\lambda_{M-1} \leq \alpha$ guarantees exact recovery.*

We provide detailed proofs and derivations in Appendix B, where we analyze the spectral decomposition's efficacy and demonstrate its suitability for multiscale molecular modeling.

### 3.4 Neural ODEs in the Spectral Domain

We propose a novel approach that models the temporal evolution of the **Fourier coefficients** using Neural ODEs. By representing molecular interactions in the frequency domain, this approach enables the decomposition of dynamics across different spatial scales and provides a more compact representation for modeling temporal evolution. Neural ODEs then learn the dynamics of these Fourier coefficients over time, offering a continuous-time framework that captures multiscale spatial and temporal interactions simultaneously.

**NeuralODE Preliminaries.** Neural Ordinary Differential Equations (Neural ODEs) (Chen et al., 2018) provide a framework for modeling continuous-time dynamics by learning the evolution of a system as a set of differential equations parameterized by a neural network. Specifically, for a system's state $\mathbf{h}(t)$, its temporal evolution is defined as:

$$\frac{d\mathbf{h}(t)}{dt} = f_\theta(\mathbf{h}(t), t), \tag{11}$$

where $f_\theta$ is a neural network parameterized by $\theta$ that learns the dynamics of $\mathbf{h}(t)$ over time. Given an initial state $\mathbf{h}(t_0)$, the state at any future time $t$ is computed by solving the ODE:

$$\mathbf{h}(t) = \mathbf{h}(t_0) + \int_{t_0}^{t} f_\theta(\mathbf{h}(\tau), \tau) d\tau. \tag{12}$$

This integral can be evaluated numerically using adaptive ODE solvers, such as Runge-Kutta methods, allowing Neural ODEs to handle irregularly sampled data and continuously model temporal dynamics.

**Joint Scalar-Vector Block-Diagonal Formulation.** We update the spectral coefficients $[\tilde{\mathbf{H}}, \tilde{\mathbf{Z}}]$ jointly. We can write the combined spectral features as

$$\begin{pmatrix} \tilde{\mathbf{H}} \\ \tilde{\mathbf{Z}} \end{pmatrix} \in \mathbb{R}^{N \times F} \oplus \mathbb{R}^{N \times m \times 3}, \tag{13}$$

and define Neural ODE dynamics over continuous time $t$:

$$\frac{d}{dt} \begin{pmatrix} \tilde{\mathbf{H}}(t) \\ \tilde{\mathbf{Z}}(t) \end{pmatrix} = \begin{pmatrix} f_\theta(\tilde{\mathbf{H}}(t), t) \\ g_\theta(\tilde{\mathbf{Z}}(t), t) \end{pmatrix}. \tag{14}$$

Mathematically, in the Fourier space, this equates to a block-diagonal operator $\mathbf{M}_\theta$. Let $\tilde{\mathbf{H}}_\omega$ and $\tilde{\mathbf{Z}}_\omega$ denote the coefficients for each frequency mode $\omega$. A single ODE step can be represented as:

$$\begin{pmatrix} \tilde{\mathbf{H}}_\omega \\ \tilde{\mathbf{Z}}_\omega \end{pmatrix} \mapsto \begin{bmatrix} \mathbf{M}_\theta^{(\mathbf{h})} & \mathbf{0} \\ \mathbf{0} & \mathbf{M}_\theta^{(\mathbf{z})} \end{bmatrix} \cdot \begin{pmatrix} \tilde{\mathbf{H}}_\omega \\ \tilde{\mathbf{Z}}_\omega \end{pmatrix}. \tag{15}$$

Here, $\mathbf{M}_\theta^{(\mathbf{h})}$ operates on the scalar channels and $\mathbf{M}_\theta^{(\mathbf{z})}$ on the vector channels, allowing them to evolve in a coordinated but distinct manner. This is beneficial to preserving the 3D interactions while keeping track of the nuanced representations in the latent space – thus preserving the 3D interactions while capturing the nuanced latent representations in a block-diagonal design.

Implementation wise, each ODE function $f_\theta$ or $g_\theta$ takes the current spectral modes ($\tilde{\mathbf{H}}_\omega$ or $\tilde{\mathbf{Z}}_\omega$) and the time $t$, and produces their rate of change for the integrator. Specifically, we use multi-head self-attention across these modes so that each frequency mode can attend to others. Formally, if $\mathbf{h}_\omega(t)$ denotes the coefficients corresponding to frequency mode $\omega$ at time $t$, the self-attention step can be written as

$$\mathbf{h}'_\omega(t) = \text{MHAttn}\big(\mathbf{h}_\omega(t)\big), \tag{16}$$

where MHAttn is the multi-head attention layer across different frequency modes. Next, the time $t$ is directly concatenated along the feature dimension via a learned embedding $\gamma(t) \in \mathbb{R}^d$, yielding

$$\mathbf{h}''_\omega(t) = \big[\mathbf{h}'_\omega(t) \,\|\, \gamma(t)\big]. \tag{17}$$

Finally, each mode $\mathbf{h}''_\omega(t)$ is transformed by a mode-wise linear weight $\mathbf{W}_\omega$, resulting in the derivative:

$$f_\theta\big(\mathbf{h}_\omega(t), t\big) \; = \; \mathbf{W}_\omega \, \mathbf{h}''_\omega(t). \tag{18}$$

A similar procedure applies for $g_\theta$, handling any additional vector channels by suitably reshaping and performing attention across modes.

To predict at time $t_k$, we numerically integrate $\frac{d}{dt}[\tilde{\mathbf{H}}, \tilde{\mathbf{Z}}]$ from $t_0$ to $t_k$ (e.g., via dopri5). Low-frequency components capture long-range, slower dynamics, while high-frequency components capture faster local fluctuations.

**Theoretical connection between spatial and temporal modes.** We also use a simplified heat equation dynamics model to demonstrate the connection of spatial GFT modes to distinct temporal dynamics.

**Proposition 3.3** (Heat-Equation Mode Dynamics). *Let $\mathcal{G} = (V, E)$ be a graph with normalized Laplacian L admitting*

$$L = U \Lambda U^\top, \quad \Lambda = \mathrm{diag}(\lambda_0, \ldots, \lambda_{N-1}), \quad 0 = \lambda_0 \le \lambda_1 \le \cdots \le \lambda_{N-1}.$$

*Consider the graph-heat evolution on a scalar signal $f(t) \in \mathbb{R}^N$:*

$$\frac{d\,f(t)}{dt} \; = \; -\,L\,f(t)\,. \tag{19}$$

*Write the kth Graph Fourier coefficient as $\alpha_k(t) \; = \; u_k^\top\, f(t)$. Then each mode evolves independently:*

$$\frac{d\,\alpha_k(t)}{dt} = -\,\lambda_k\,\alpha_k(t), \tag{20}$$

*with closed-form solution $\alpha_k(t) \; = \; \alpha_k(0) \exp\big(-\lambda_k\, t\big)$. In particular, modes with $\lambda_k$ small decay* slowly *(long time-scales), while modes with $\lambda_k$ large decay* rapidly *(short time-scales).*

Although our GF-NODE dynamics are *learned* rather than the pure heat equation, Proposition 3.3 shows that under any diffusion-like operator the Laplacian eigenvalues $\lambda_k$ set intrinsic time-scales for each mode. The detailed proof and theoretical analysis can be found in Appendix C.

### 3.5 From Spectral Domain to Physical Domain

**Inverse GFT.** After evolving the spectral coefficients $\tilde{\mathbf{H}}(t_k)$ and $\tilde{\mathbf{Z}}(t_k)$, we recover the node-level signals via the inverse GFT:

$$\mathbf{H}(t_k) = \mathbf{U}_{(:,0:M)}\tilde{\mathbf{H}}(t_k), \quad \mathbf{Z}_c(t_k) = \mathbf{U}_{(:,0:M)}\tilde{\mathbf{Z}}(t_k), \tag{21}$$

where $\mathbf{U}_{(:,0:M)}$ (the matrix of the first $M$ eigenvectors) is the same truncated basis used during the forward GFT. Finally, we restore the global translation by adding back the mean $\overline{\mathbf{Z}}$ that was subtracted earlier:

$$\mathbf{Z}(t_k) = \mathbf{Z}_c(t_k) + \overline{\mathbf{Z}}. \tag{22}$$

**Graph Neural Network Decoder.** We refine local interactions at each predicted time $t_k$ with a GNN decoder that again operates on $\big[\mathbf{H}(t_k), \mathbf{Z}(t_k)\big]$. This step can help capture short-range correlations that may not be fully resolved in the spectral update. The decoder GNN has a structure similar to the encoder:

$$\mathbf{m}_{ij}^{(\mathrm{dec})} = \phi'_e\Big(\mathbf{h}_i(t_k),\, \mathbf{h}_j(t_k),\, \mathbf{r}_{ij}^{(\mathrm{dec})}(t_k)\Big), \tag{23}$$

$$\mathbf{h}_i^{(\mathrm{dec})}(t_k) = \phi'_h\Big(\mathbf{h}_i(t_k),\, \sum_{j \in \mathcal{N}(i)} \mathbf{m}_{ij}^{(\mathrm{dec})}\Big), \tag{24}$$

$$\mathbf{x}_i^{(\mathrm{dec})}(t_k) = \mathbf{x}_i(t_k) \; + \; \frac{1}{|\mathcal{N}(i)|} \sum_{j \in \mathcal{N}(i)} \psi'\big(\mathbf{m}_{ij}^{(\mathrm{dec})}\big), \tag{25}$$

and similarly for $\mathbf{v}_i$. Here, $\mathbf{r}_{ij}^{(\mathrm{dec})}(t_k) = \mathbf{x}_i(t_k) - \mathbf{x}_j(t_k)$, and $\phi'_e$, $\phi'_h$, $\psi'$ are learnable functions.

## 4 Experiments

In this section, we provide a comprehensive evaluation of our proposed model on molecular dynamics datasets, comparing its performance against several baselines and conducting detailed ablation studies to assess the contributions of the different components. Our experimental evaluation is designed to address the following key **research questions**:

- **RQ1: Prediction Accuracy.** Does the proposed GF-NODE framework deliver improved molecular dynamics prediction accuracy compared to state-of-the-art methods?
- **RQ2: Continuous Time Modeling.** How effectively does the continuous-time evolution component capture multiscale temporal dynamics—including long-horizon forecasting and super-resolution—compared to variants without continuous-time propagation?
- **RQ3: Spatial Multiscale Modeling.** How crucial is the explicit spectral decomposition for capturing spatial multiscale interactions, and how do different Fourier mode interaction schemes and the number of retained modes affect overall performance?
- **RQ4: Theoretical Justification.** Can we rigorously establish the correspondence between spatial frequency modes (graph Laplacian eigenvectors) and temporal dynamics scales, providing theoretical foundation for the GF-NODE architecture?

### 4.1 Dataset

We evaluate our model using both the Revised MD17 dataset (Christensen & von Lilienfeld, 2020) and the original MD17 dataset (Chmiela et al., 2017), which contain molecular dynamics trajectories for small molecules including Aspirin, Benzene, Ethanol, Malonaldehyde, Naphthalene, Salicylic Acid, Toluene, and Uracil. The datasets provide atomic trajectories simulated under quantum mechanical forces, capturing realistic molecular motions. To demonstrate scalability, we further evaluate on five larger molecular systems with 20–326 heavy atoms: alanine dipeptide ($Ala_2$) from MDShare, Ac-Ala$_3$-NHMe, AT-AT-CG-CG, Bucky-Catcher, and a double-walled carbon nanotube, from the MD22 dataset (Chmiela et al., 2023). These larger systems exhibit complex multiscale dynamics spanning local bond vibrations to global conformational changes.

We also evaluated our model on the alanine dipeptide dataset (Schreiner et al., 2024), a standard benchmark for studying conformational dynamics in proteins. Our task is to predict the future positions of atoms given the initial state of the molecular system.

### 4.2 Experimental Setup

For each molecule, we partition the trajectory data into training, validation, and test sets, using 500 samples for training, 2000 for validation, and 2000 for testing. The time scope $\Delta T$ for each piece of data is set to 3000 simulation steps, providing a challenging prediction horizon. To demonstrate long-term stability, we extend our evaluation to $\Delta t = 10{,}000$ steps and up to 15,000 MD steps for comprehensive temporal analysis.

A key aspect of our experimental setup is the use of **irregular timestep sampling**, in contrast to the equi-timestep sampling used in some baseline models like EGNO, to better mimic the variable time intervals in real-world physical systems. This setting tests the models' ability to handle irregular temporal data. Although each trajectory spans 3000 simulation steps, we randomly sample only 8 data points per instance. Because these samples are drawn from different points in the 3000-step window across data instances, this strategy enables the model to learn the dynamics over the entire time span without requiring training on every timestep, thereby significantly enhancing **efficiency**. Nevertheless, we also provide evaluations based on equi-timestep sampling in Appendix E for completeness.

### 4.3 Baseline Models

We compare our model against several state-of-the-art approaches:

- **NDCN** (Zang & Wang, 2020): A Graph Neural ODE model that integrates graph neural networks into the ODE framework to learn continuous-time dynamics of networked systems.
- **LG-ODE** (Huang et al., 2020): A latent graph-based ODE model that integrates latent representations into continuous-time evolution.

- **EGNN** (Satorras et al., 2021): An Equivariant Graph Neural Network that models molecular systems using 3D equivariant message passing but without explicit time propagation.
- **EGNO** (Xu et al., 2024): An Equivariant Graph Neural Operator that captures temporal dynamics using neural operators with regular timesteps.
- **ITO** (Schreiner et al., 2024): An Implicit Time-stepping Operator that integrates differential equations into the learning process for temporal evolution.

These baselines represent a range of approaches for modeling molecular dynamics, including methods that focus primarily on spatial modeling (EGNN) or temporal modeling (NDCN, LG-ODE, EGNO, ITO).

## 4.4 Results and Analysis (RQ1)

*Units and Calculations:* Note that the alanine dipeptide dataset operates in nanometers (nm), whereas the MD17 dataset uses angstroms (Å). When calculating bond lengths and bond angles, we consider all heavy atoms (excluding hydrogen) to focus on the core structure.

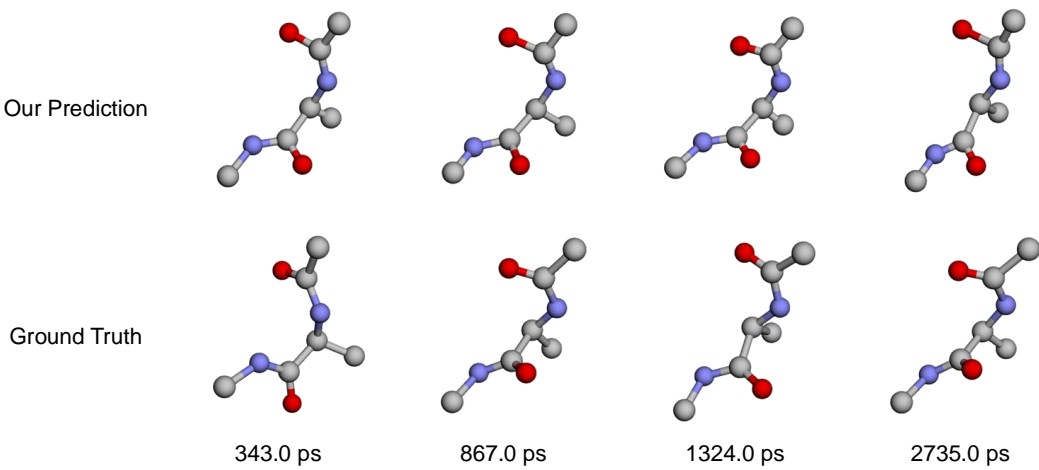

Figure 4: Representative snapshots of alanine dipeptide, comparing our model's predicted conformations (top row) against the ground-truth simulation (bottom row) at four different timestamps. The close agreement illustrates the model's ability to preserve key structural features over an extended trajectory.

To address **RQ1**, we evaluate the performance of our model and the baselines on both the Revised MD17 dataset and larger molecular systems, with Figure 4 providing visualization of our predictions. Our comprehensive evaluation includes results at multiple time horizons ($\Delta t = 3000$ and 10,000 steps) and extended evaluations up to 15,000 MD steps. The test Mean Squared Error (MSE) for our model and the baseline methods are summarized in Tables 1 and 3 for the original evaluation, under **irregular** timestep sampling. Comprehensive results on the Revised MD17 dataset and larger molecular systems can be found in Tables 11–14 in the appendix, demonstrating consistent improvements across all systems and time horizons. For completeness, we also provide results in the equi-timestep setting in Table 7 in Appendix E.

Table 1: MSE ($\times 10^{-2}$ Å$^2$) on the MD17 dataset with **irregular** timestep sampling. Best results are in **bold**, and second-best are underlined.

|  | Aspirin | Benzene | Ethanol | Malonaldehyde | Naphthalene | Salicylic | Toluene | Uracil |
|---|---|---|---|---|---|---|---|---|
| NDCN | 29.75±0.02 | 70.13±0.98 | 10.05±0.02 | 42.28±0.07 | 2.30±0.00 | 3.43±0.05 | 12.33±0.00 | 2.39±0.00 |
| LG-ODE | 51.65±0.01 | 68.29±0.21 | 12.32±0.05 | 43.95±0.07 | 2.38±0.02 | 2.85±0.08 | 18.11±0.09 | 2.38±0.07 |
| EGNN | 9.09±0.10 | 49.15±1.68 | 4.46±0.01 | 12.52±0.05 | 0.40±0.02 | 0.89±0.01 | 8.98±0.09 | 0.64±0.00 |
| EGNO | 10.60±0.01 | 52.53±2.40 | 4.52±0.06 | 12.89±0.06 | 0.46±0.01 | 1.07±0.00 | 9.31±0.10 | 0.67±0.01 |
| ITO | 12.74±0.10 | 57.84±0.86 | 7.23±0.00 | 19.53±0.01 | 1.77±0.01 | 2.53±0.03 | 9.96±0.04 | 1.71±0.15 |
| Ours | **6.46**±0.03 | **1.52**±0.08 | **2.74**±0.05 | **10.54**±0.01 | **0.23**±0.02 | **0.63**±0.01 | **1.80**±0.05 | **0.41**±0.01 |

Table 2: Mean Absolute Errors (MAEs) and relative errors for bond lengths and bond angles on alanine dipeptide. Best results are in **bold**.

| Model | Bond Length MAE (nm) | Rel. Err. (%) | Bond Angle MAE (°) | Rel. Err. (%) |
|---|---|---|---|---|
| EGNN | $0.0209 \pm 0.0006$ | $15.32 \pm 0.49$ | $12.44 \pm 0.91$ | $10.48 \pm 0.76$ |
| EGNO | $0.0229 \pm 0.0018$ | $16.75 \pm 1.23$ | $10.54 \pm 0.11$ | $8.89 \pm 0.11$ |
| **Ours** | $\mathbf{0.0188} \pm 0.0022$ | $\mathbf{13.74} \pm 1.66$ | $\mathbf{10.47} \pm 1.03$ | $\mathbf{8.80} \pm 0.89$ |

From Table 1, we observe that our model consistently outperforms the baseline methods across all eight molecules under irregular timestep sampling. This demonstrates the effectiveness of our approach in jointly modeling spatial and temporal multiscale interactions. The performance gains are particularly pronounced for molecules such as Benzene and Aspirin, where our model significantly reduces the MSE compared to the baselines. Our comprehensive evaluation on the Revised MD17 dataset and five larger molecular systems (20–326 heavy atoms) shows that GF-NODE achieves the best accuracy on *all* cases across multiple time horizons ($\Delta t = 3000$ and $10,000$ steps),

Table 3: Mean Squared Error (MSE) ($\times 10^{-3}$ nm$^2$) on the alanine dipeptide dataset. Best results are in **bold**.

| Model | MSE ($\times 10^{-3}$ nm$^2$) |
|---|---|
| NDCN | $12.27 \pm 0.19$ |
| ITO | $26.95 \pm 0.19$ |
| EGNO | $6.92 \pm 0.26$ |
| EGNN | $5.67 \pm 0.08$ |
| **Ours** | $\mathbf{4.48} \pm 0.07$ |

with detailed results provided in the appendix (Tables 11–14). The model demonstrates superior long-horizon stability up to 15,000 MD steps (Figures 10 and 11), capturing both short-range covalent vibrations and long-range collective modes such as radial breathing in carbon nanotubes.

**Benzene Drifting.** Interestingly, we find that Benzene exhibits substantial drift during simulation. In our data, the maximum $x$-coordinate is 197.981 Å, while the minimum $x$-coordinate is -178.112 Å, indicating large translations and rotations of the entire molecule. In contrast, other molecules exhibit minimal net translation, typically remaining within ±3Å from the origin. Methods that enforce strict invariance to translations and rotations (e.g., EGNN) may underperform on such drifting systems, since the global drift is part of the actual dynamics. Indeed, we discovered that *replacing EGNN-based layers with standard message passing layers* (e.g., SAGEConv) *can further boost performance on drifting molecules* like Benzene. We provide details of the dataset statistics in Table 6 of Appendix A, and experiment results using different GNN architectures in Table 9 of Appendix E.

**Experimental Results on Alanine Dipeptide.** In addition, from Table 3, our model achieves the lowest MSE among all compared methods. This indicates a significant improvement on more complex molecular dynamics data and demonstrates the robustness of our approach.

**Analysis of Molecular Structure Recovery.** To further evaluate predictions at a structural level, we analyzed the bond lengths and bond angles for alanine dipeptide. Table 2 shows the Mean Absolute Errors (MAEs) and relative errors for bond lengths and angles. Our method achieves the lowest errors, indicating superior recovery of internal molecular structures compared to baselines. Additionally, extensive radial distribution function (RDF) analysis for five representative systems (Figures 14–17 in Appendix F) confirms that GF-NODE accurately reproduces both first-shell peaks and longer-range oscillations of the *ab initio* reference, demonstrating that predicted trajectories maintain realistic structural correlations well beyond pairwise coordinate accuracy.

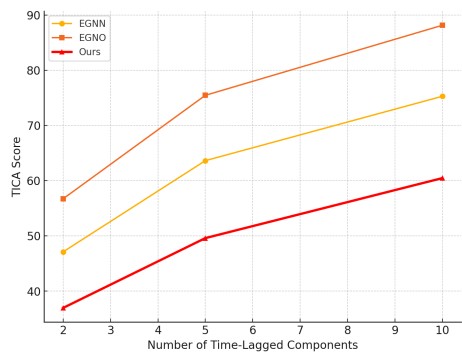

Figure 5: TICA scores for varying numbers of components. Lower scores indicate better alignment with slow modes.

**TICA Analysis.** Time-lagged Independent Component Analysis (TICA)(Molgedey & Schuster, 1994) is used to extract slow collective motions from the trajectories. Figure 5 shows that, across various numbers

of TICA components, our model consistently achieves lower TICA scores (i.e., better alignment with the underlying slow modes) compared to EGNN and EGNO, indicating more effective capture of the underlying multiscale dynamics.

### 4.5 Continuous-Time Dynamics and Temporal Super-Resolution (RQ2)

**Ablation on continuous time components.** To assess our model's ability to capture temporal multiscale dynamics, we perform an ablation study on the continuous-time components. Specifically, we investigate:

1. **No ODEs Evolution:** Removing the continuous-time evolution altogether.
2. **No ODEs on Scalar Channels:** Freeze the ODEs for among scalar features **h** during time propagation.
3. **No ODEs on Vector Channels:** Freeze the ODEs among vector features **x** during time propagation.

Table 4 reports the test errors (MSE) for these variants across several molecules. The inferior performance of the ablated models confirms the importance of the continuous-time dynamics and the effectiveness of the block-diagonal architecture that jointly propagates scalar and vector features.

Table 4: Ablation study on continuous-time evolution components (MSE $\times 10^{-2}$ Å$^2$ values). Lower values indicate better performance. All results are **inferior** to our standard model (w/ ode).

|            | Aspirin         | Benzene         | Ethanol         | Malonaldehyde    | Naphthalene     | Salicylic       | Toluene         | Uracil          |
| ---------- | --------------- | --------------- | --------------- | ---------------- | --------------- | --------------- | --------------- | --------------- |
| no_ode     | 6.56±0.03       | 1.87±0.08       | 3.09±0.05       | 11.58±0.02       | 0.42±0.02       | 0.86±0.01       | 3.06±0.05       | 0.61±0.01       |
| no_ode_h   | 6.77±0.05       | 1.89±0.06       | 3.11±0.03       | 10.70±0.02       | 0.42±0.01       | 0.88±0.02       | 3.48±0.04       | 0.59±0.02       |
| no_ode_x   | 6.95±0.03       | 1.79±0.07       | 3.74±0.05       | 10.61±0.03       | 0.42±0.01       | 0.87±0.01       | 2.89±0.04       | 0.59±0.01       |
| w/ ode     | **6.46**±0.03   | **1.52**±0.08   | **2.74**±0.05   | **10.54**±0.01   | **0.23**±0.02   | **0.63**±0.01   | **1.80**±0.05   | **0.41**±0.01   |

**Super-Resolution Task.** To further validate temporal generalization, we perform a super-resolution experiment on the alanine dipeptide dataset by predicting trajectories at a $10\times$ finer temporal resolution than the training samples, under the equi-timestep setting. Figure 6 compares the MSE for the original versus the super-resolved predictions. Our model maintains low error under super-resolution, demonstrating its ability to interpolate continuous dynamics effectively.

**Long-Term Prediction Stability.** We evaluate the stability of long-term predictions across extended time horizons. We test models at 1000, 2000, 3000, 4000, and 5000 simulation steps on two representative molecules (Benzene and Malonaldehyde). Figure 7 illustrates how errors evolve over these extended horizons. Our model maintains superior performance with slower error growth, indicating better capture of global low-frequency dynamics essential for accurate long-term predictions. In contrast, baselines lacking explicit multiscale modeling accumulate errors more rapidly. Extended evaluations up to 15,000 MD steps on both the Revised MD17 dataset (Figure 10) and larger molecular systems (Figure 11) further demonstrate the superior long-horizon stability of our approach, with computational efficiency analysis provided in Figure 12.

Table 4 in Appendix E provides additional ablations on the different types of temporal embeddings used in the ODE functions. A simple concatenation of the timestamp would work well enough.

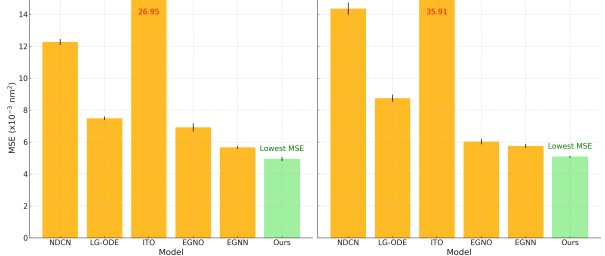

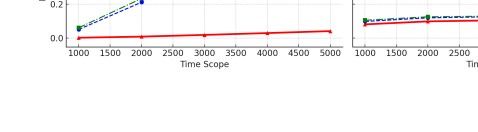

Figure 6: Comparison of MSE on the alanine dipeptide dataset: Original vs. $10\times$ Super-resolution.

Figure 7: Error growth over long-term forecasts (up to 5000 steps) for Benzene and Malonaldehyde. Models with too high an error not presented here.

### 4.6 Spatial Frequency Decomposition and Multiscale Analysis (RQ3)

To examine the role of spatial multiscale modeling, we perform ablations on the spectral decomposition and mode interaction components. We consider two sets of modifications:

1. **Ablation of the Spatial Decomposition:**
   - *No Fourier-based Decomposition:* The model is run without any frequency-based transformation. The spectral decomposition is replaced by a multi-layer perception.
   - *Replacing GFT with FFT:* The Graph Fourier Transform is substituted with a standard Fast Fourier Transform, disregarding the graph structure.
2. **Fourier Mode Interaction Schemes:** We compare different strategies for inter-mode communication:
   - *Attention-based Interaction:* Each Fourier mode interacts with others via a multi-head self-attention mechanism (Used in our standard model).
   - *Concatenation-based Interaction:* Modes are concatenated before being processed.
   - *No Interaction:* Each mode is propagated independently.

Table 5 compiles the results for these spatial ablations. We observe that using the GFT for spectral decomposition—rather than an MLP or FFT—is most effective, underscoring the importance of capturing the inherent graph structure. Moreover, interaction schemes (whether via attention or concatenation) improve performance over treating modes independently.

Table 5: MSE ($\times 10^{-2}$ Å$^2$) on the MD17 dataset with **irregular** timestep sampling for variants in spatial decomposition and Fourier modes interactions. Best results are in **bold**, the standard model (GFT & Attn.).

|  | Aspirin | Benzene | Ethanol | Malonaldehyde | Naphthalene | Salicylic | Toluene | Uracil |
|---|---|---|---|---|---|---|---|---|
| FFT | 6.53±0.03 | 1.94±0.08 | 3.36±0.05 | 10.83±0.01 | 0.43±0.02 | 0.88±0.01 | 3.36±0.05 | 0.60±0.01 |
| No Fourier | 6.63±0.04 | 1.99±0.10 | 3.28±0.06 | 10.68±0.07 | 0.43±0.02 | 0.88±0.01 | 3.73±0.09 | 0.60±0.05 |
| No Interaction | 6.51±0.03 | 1.78±0.08 | 3.28±0.05 | 10.64±0.01 | 0.41±0.02 | 0.87±0.02 | 3.08±0.05 | 0.59±0.01 |
| Concat. | 6.55±0.03 | 1.76±0.07 | 3.13±0.05 | 10.58±0.01 | 0.43±0.02 | 0.88±0.01 | 2.80±0.05 | 0.58±0.01 |
| GFT & Attn. | **6.46**±0.03 | **1.52**±0.08 | **2.74**±0.05 | **10.54**±0.01 | **0.23**±0.02 | **0.63**±0.01 | **1.80**±0.05 | **0.41**±0.01 |

**Impact of the Number of Fourier Modes.** Finally, we investigate how the number of retained Fourier modes affects performance. Figure 8 plots the MSE against different numbers of modes used in the spectral decomposition. We observe that the performance improves as more modes are included up to a threshold, beyond which additional modes yield diminishing returns. This behavior is consistent with the theoretical analysis presented earlier in Theorem 3.2. The number of modes used to get the optimal results for each type of molecule can be found in Table 6 of Appendix A.

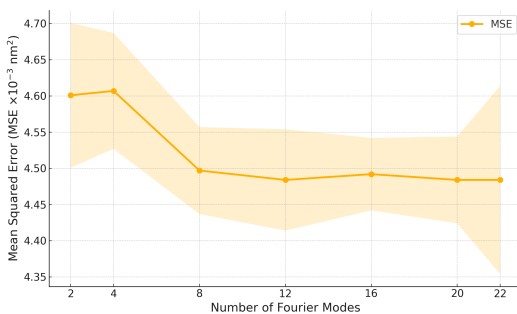

Figure 8: Effect of the number of Fourier modes on prediction performance. Performance plateaus beyond $n = 12$.

### 4.7 Theoretical Justification and Spatial-Temporal Correspondence (RQ4)

To address concerns about the conceptual rigor of linking spatial GFT modes to distinct temporal dynamics, we provide both theoretical insight and comprehensive empirical validation. Our analysis demonstrates that spatial frequency decomposition naturally corresponds to different temporal evolution scales in molecular systems.

**Theoretical Insight through Simplified Model.** We establish theoretical foundation through heat equation analysis on a simplified diffusion model (detailed in Appendix C). This analysis shows that on a continuous domain, the graph Laplacian eigenvalues $\lambda_k$ intrinsically determine the temporal evolution rates of different spatial frequency modes, with higher eigenvalues corresponding to faster decay rates. While this provides valuable theoretical insight, it represents a simplified model of molecular dynamics.

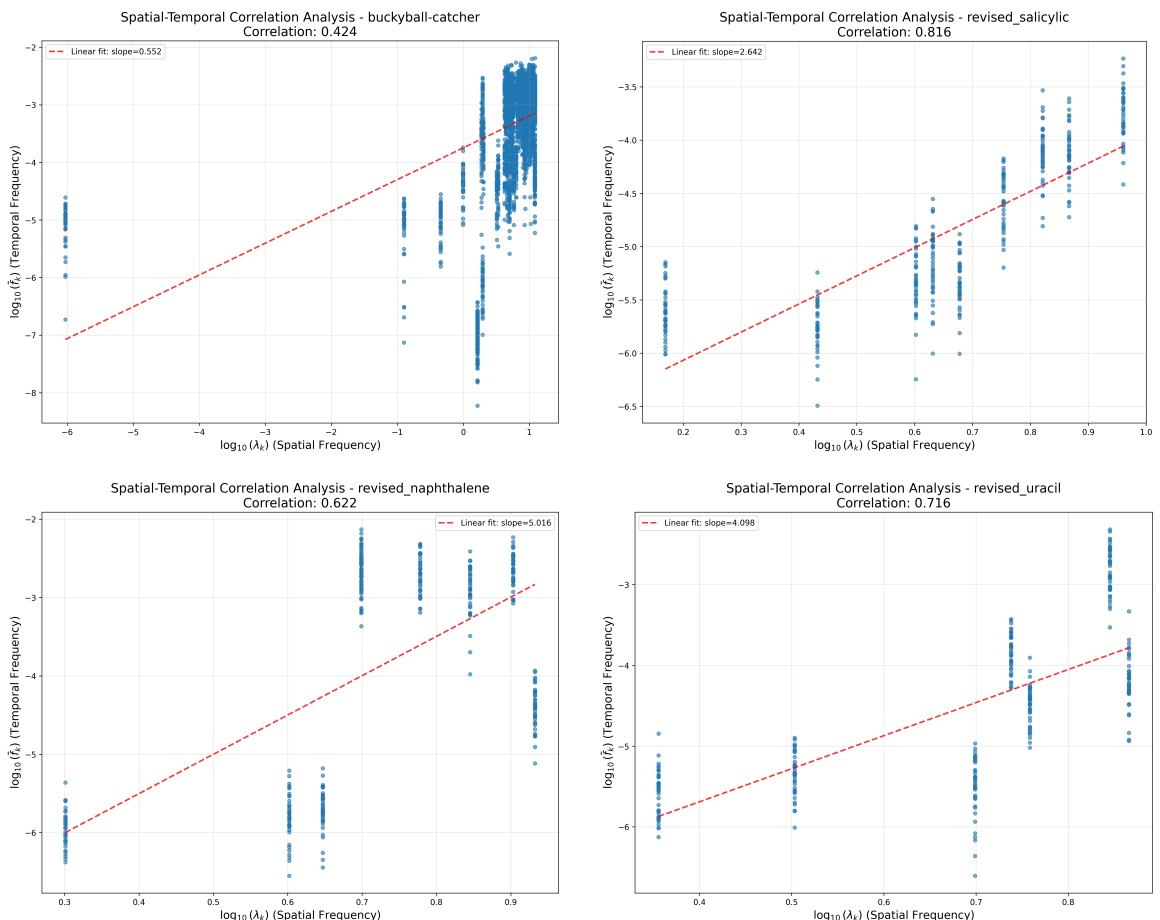

Figure 9: Spatial-temporal correlation analysis across four molecular systems: (a) Buckyball-catcher, $r = 0.424$, $\alpha = 0.552$; (b) Salicylic acid, $r = 0.816$, $\alpha = 2.642$; (c) Naphthalene, $r = 0.622$, $\alpha = 5.016$; (d) Uracil, $r = 0.716$, $\alpha = 4.098$. Each panel shows a log-log plot of characteristic temporal frequency $\bar{f}_k$ versus spatial frequency $\lambda_k$ (graph Laplacian eigenvalues) for individual eigenmodes. The consistently positive correlations validate the predicted correspondence between spatial and temporal scales across diverse molecular architectures.

**Empirical Validation on Real Molecular Systems.** To validate this correspondence on actual molecular trajectories, we developed a comprehensive Theoretical Framework for Joint Spatial-Temporal Analysis in Appendix Section C.1that quantifies spatial-temporal correlations across diverse molecular systems. Our empirical results on five representative systems (Naphthalene, Salicylic Acid, Uracil, Bucky-Catcher, and double-walled nanotube), as demonstrated in Figure 9 shows clear quantitative relationships between spatial frequency modes and temporal dynamics scales.

Key findings include: (1) Distinct spatial modes exhibit different temporal autocorrelation decay rates, with low-frequency modes showing slower decay (long-term dynamics) and high-frequency modes showing faster decay (short-term fluctuations). (2) Cross-correlation analysis reveals that spatial modes with similar frequencies exhibit stronger temporal correlations. (3) The empirical correspondence between spatial eigenvalues and temporal scales aligns with theoretical predictions from the simplified heat equation model.

These results provide strong empirical support for our core hypothesis that spatial spectral decomposition effectively separates distinct temporal dynamics scales, justifying the GF-NODE architecture's design principle of processing different frequency modes with tailored temporal evolution pathways.

## 5 Conclusion

In this work, we presented Graph Fourier Neural ODEs (GF-NODE), a novel framework that unifies spatial spectral decomposition with continuous-time evolution to effectively model the multiscale dynamics inherent in molecular systems. By decomposing molecular graphs via the Graph Fourier Transform, our approach explicitly disentangles global conformational changes from local vibrational modes, and the subsequent Neural ODE-based propagation enables flexible, continuous-time forecasting of these dynamics.

Our extensive evaluations span benchmark datasets including the Revised MD17 dataset, five larger molecular systems (20–326 heavy atoms), and extended temporal horizons up to 15,000 MD steps. GF-NODE achieves state-of-the-art performance across *all* evaluated systems and time horizons, demonstrating superior long-horizon trajectory prediction while accurately preserving essential molecular geometries. Comprehensive structural analyses including radial distribution functions confirm that our model maintains realistic molecular correlations well beyond coordinate-level accuracy. The ablation studies further highlight the critical role of both the spectral decomposition and the continuous-time dynamics in capturing complex spatial-temporal interactions, with formal $SO(3)$ equivariance guarantees provided. Importantly, our theoretical insight through heat equation analysis on simplified diffusion models, combined with comprehensive empirical validation on real molecular dynamics trajectories, establishes a rigorous foundation for the spatial-temporal correspondence that underlies our approach. The empirical validation through our Theoretical Framework for Joint Spatial-Temporal Analysis demonstrates quantitative relationships between spatial frequency modes and temporal dynamics scales across diverse molecular systems, confirming that spectral decomposition effectively separates distinct temporal evolution patterns.

These findings suggest that incorporating physics-informed spectral decomposition principles into neural architectures represents a promising direction for advancing molecular dynamics modeling, with broader implications for understanding and predicting complex multiscale phenomena in chemical and biological systems.

## 6 Broader Impact Statement

Our work contributes to advancing molecular dynamics simulations, which have broad implications in scientific discovery, particularly in drug design, materials science, and biomolecular modeling. By improving the accuracy and efficiency of multiscale molecular predictions, our approach could accelerate the discovery of novel therapeutics and facilitate the design of functional materials with tailored properties.

While our method relies on data-driven modeling, it does not replace physics-based simulations but rather augments them, reducing computational costs while maintaining interpretability. As with any machine learning-driven approach in scientific domains, care must be taken to ensure model reliability, particularly in high-stakes applications such as drug development. Further validation and collaboration with domain experts will be essential to maximize the positive societal impact of our work while mitigating risks related to model uncertainty.

## 7 Acknowledgement

This work was partially supported by NSF 2211557, NSF 2119643, NSF 2303037, NSF 2312501; the NAIRR Pilot Program (NSF 2202693, NSF 2312501); the SRC JUMP 2.0 Center; Amazon Research Awards; Snapchat Gifts; and C-CAS (Center for Computer Assisted Synthesis, NSF CHE–2202693).

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

## A    Experiment Setup

In this section, we describe key details about the datasets used (MD17 and alanine dipeptide), including the simulation step sizes, and then outline the hyperparameter choices and training procedures.

### A.1    Datasets

We use the MD17 dataset (Chmiela et al., 2017) for small-molecule dynamics, and the alanine dipeptide dataset (Schreiner et al., 2024) for conformational analysis in proteins. Statistics for MD17 are provided in Table 6. From the table, we observe Benzene has a very marked drifting that is much larger that the scale of the molecule itself. The MD17 dataset was generated using ab initio molecular dynamics (MD) simulations with a time step of 0.5 femtoseconds (fs). The simulations were conducted in the NVT ensemble at 500 K for a total duration of 2000 picoseconds (ps). The final dataset was created by subsampling the full trajectory, preserving the Maxwell-Boltzmann distribution for the energies. For the alanine dipeptide dataset, we use the protein fragment's trajectory recorded at a step size of 1.0 ps.

Table 6: Summary statistics for molecular structures, including the number of atoms, position extrema ($X_{min}$, $X_{max}$, $X_{mean}$), and velocity extrema ($V_{min}$, $V_{max}$, $V_{mean}$). The numbers of modes used to get the optimal results are also listed.

| Molecule | #Atoms | #Modes used | $X_{min}$ | $X_{max}$ | $X_{mean}$ | $V_{min}$ | $V_{max}$ | $V_{mean}$ |
|---|---|---|---|---|---|---|---|---|
| benzene | 6 | 4 | $-178.112$ | 197.981 | $-27.737$ | $-0.004$ | 0.003 | $-0.000$ |
| aspirin | 13 | 6 | $-3.720$ | 3.105 | 0.026 | $-0.011$ | 0.012 | 0.000 |
| ethanol | 3 | 3 | $-1.398$ | 1.417 | $-0.004$ | $-0.011$ | 0.010 | $-0.000$ |
| malonaldehyde | 5 | 5 | $-2.397$ | 2.370 | 0.000 | $-0.010$ | 0.009 | 0.000 |
| naphthalene | 10 | 4 | $-2.597$ | 2.593 | $-0.000$ | $-0.012$ | 0.011 | 0.000 |
| salicylic | 10 | 8 | $-2.734$ | 2.581 | $-0.051$ | $-0.013$ | 0.012 | $-0.000$ |
| toluene | 7 | 7 | $-1.990$ | 2.630 | $-0.015$ | $-0.010$ | 0.012 | 0.000 |
| uracil | 8 | 6 | $-2.338$ | 2.558 | 0.012 | $-0.012$ | 0.011 | 0.000 |

### A.2    Training Setup

**Hyperparameters.** We train all models using the Adam optimizer at a learning rate of $1 \times 10^{-4}$ and apply a weight decay of $1 \times 10^{-15}$ for regularization. Each experiment runs for 5000 epochs, processing batches of size 50 at each training step. For molecular trajectory prediction, we set the sequence length (the number of timesteps) to 8, meaning each training sample contains 8 frames from the overall simulation. In our main experiments, we fix the hidden feature dimensionality to 64 and use a maximum future horizon of 3000 simulation steps when constructing training samples. We also specify the dopri5 ODE solver with relative and absolute tolerances of $1 \times 10^{-3}$ and $1 \times 10^{-4}$, respectively, to integrate the continuous-time model components. Although many of these choices (e.g., total layers, solver tolerances) can be altered, we find these particular settings maintain a good balance of accuracy and computational efficiency.

**Training Procedure.** During training, each mini-batch is formed by sampling short segments of length 8 from the molecule's dynamics trajectory. The model then predicts future positions of atoms after continuous-time evolution, and the Mean Squared Error (MSE) between predicted coordinates and ground-truth coordinates is minimized. We checkpoint models whenever validation performance improves, and at the end of training, we report results using the best-performing checkpoint according to the validation set. In addition, regular evaluations on the test set help track the model's generalization to unseen trajectories. The overall process can be summarized as: (1) load training samples, (2) form mini-batches of molecular frames, (3) perform forward pass through the model to generate predictions, (4) compute the MSE loss, (5) update model parameters via backpropagation, repeating for each epoch until convergence.

By following these procedures with our chosen hyperparameters, we have observed stable convergence across both MD17 and alanine dipeptide datasets, as well as strong generalization to different segments of the trajectory during test runs.

### A.3 Loss Function

To train the model, we use the Mean Squared Error (MSE) loss, which measures the difference between the predicted atomic positions and the ground truth positions at each predicted time point. Given that the goal is to predict the molecular conformations at time points $t_1, t_2, \ldots, t_K$, the MSE is calculated as follows:

Let $\mathbf{x}_i^{t_j} \in \mathbb{R}^3$ be the ground truth 3D coordinates of atom $i$ at time $t_j$, and $\tilde{\mathbf{x}}_i^{t_j} \in \mathbb{R}^3$ be the predicted coordinates for the same atom at time $t_j$. The MSE loss is defined as:

$$\mathcal{L}_{\text{MSE}} = \frac{1}{NK} \sum_{j=1}^{K} \sum_{i=1}^{N} \left\| \mathbf{x}_i^{t_j} - \tilde{\mathbf{x}}_i^{t_j} \right\|_2^2, \tag{26}$$

where $N$ is the number of atoms and $K$ is the number of time points.

This loss encourages the model to minimize the Euclidean distance between the predicted and actual atomic positions across all time steps, ensuring accurate trajectory prediction for the molecular system.

The MSE loss is applied at each time point, thus aligning the predicted future states with the true molecular dynamics trajectory. The model is trained by minimizing $\mathcal{L}_{\text{MSE}}$ over all predicted time points.

# B  Theoretical Guarantees on Spectral Decompositions

Below, we present a concise mathematical exposition on the theoretical underpinnings of the Graph Fourier Transform (GFT) decomposition used in our framework. We explain how the eigenvalue–eigenvector structure of the graph Laplacian $\mathbf{L}$ induces a decomposition of graph signals into low-frequency (global) and high-frequency (local) modes, and we justify truncating to the first $M$ modes. We follow standard nomenclature in spectral graph theory (Chung, 1997).

## B.1  Preliminaries and Definitions

**Definition B.1** (Graph Laplacian). Let $\mathcal{G} = (\mathcal{V}, \mathcal{E})$ be an undirected graph with $N = |\mathcal{V}|$ vertices. Let $\mathbf{A} \in \mathbb{R}^{N \times N}$ be its adjacency matrix, and let $\mathbf{D}$ be the diagonal degree matrix, where

$$\mathbf{D}(i,i) = \sum_{j=1}^{N} \mathbf{A}(i,j). \tag{27}$$

The *graph Laplacian* is defined as

$$\mathbf{L} = \mathbf{D} - \mathbf{A}. \tag{28}$$

It is well known that $\mathbf{L}$ is real symmetric and positive semidefinite. In fact, the eigenvalues of the Laplacian matrix are real and non-negative.

**Definition B.2** (Graph Fourier Transform (GFT)). Given the eigen-decomposition

$$\mathbf{L} = \mathbf{U}\boldsymbol{\Lambda}\mathbf{U}^{\top}, \tag{29}$$

where

$$\boldsymbol{\Lambda} = \mathrm{diag}(\lambda_0, \lambda_1, \ldots, \lambda_{N-1}), \quad 0 = \lambda_0 \leq \lambda_1 \leq \cdots \leq \lambda_{N-1}, \tag{30}$$

and $\mathbf{U} = \begin{bmatrix} \mathbf{u}_0 \mid \mathbf{u}_1 \mid \cdots \mid \mathbf{u}_{N-1} \end{bmatrix}$ stores the corresponding orthonormal eigenvectors in columns. For a graph signal

$$\mathbf{x} = \left( x_1, x_2, \ldots, x_N \right)^{\top} \in \mathbb{R}^N, \tag{31}$$

the *Graph Fourier Transform (GFT)* of $\mathbf{x}$ is given by

$$\widehat{\mathbf{x}} = \mathbf{U}^{\top}\mathbf{x}, \tag{32}$$

and the *inverse GFT* is

$$\mathbf{x} = \mathbf{U}\widehat{\mathbf{x}}. \tag{33}$$

## B.2  Truncation and Mode Selection

**Lemma B.3** (Approximation Error for Spectral Truncation). *Let $\mathbf{x} \in \mathbb{R}^N$ be any graph signal, and let $\mathbf{x}_{(M)}$ be its spectral approximation obtained by keeping the first $M$ modes. Then*

$$\|\mathbf{x} - \mathbf{x}_{(M)}\|_2^2 = \sum_{k=M}^{N-1} \left| \mathbf{u}_k^{\top}\mathbf{x} \right|^2. \tag{34}$$

*Moreover, if $\mathbf{x}$ is $\alpha$-bandlimited in the sense that*

$$\mathbf{u}_k^{\top}\mathbf{x} = 0 \quad \text{for all } \lambda_k > \alpha, \tag{35}$$

*then choosing $M$ such that $\lambda_{M-1} \leq \alpha$ yields an exact recovery $\mathbf{x} = \mathbf{x}_{(M)}$.*

*Proof.* See the main text for details. We expand the signal in the Laplacian eigenbasis $\{\mathbf{u}_k\}$, and observe that discarding all modes with $k \geq M$ removes the corresponding frequency components. □

### B.3 Low-Frequency vs. High-Frequency Modes

Because $\mathbf{L}$ is positive semidefinite and the eigenvalues $\{\lambda_i\}$ increase with $i$, smaller eigenvalues correspond to slow, global variations, while larger eigenvalues capture more oscillatory, local phenomena.

**Proposition B.4** (Global vs. Local Spatial Scales). *Let $\mathbf{u}_k$ be the $k$-th eigenvector of $\mathbf{L}$ with eigenvalue $\lambda_k$. Suppose $\mathbf{x}$ encodes atomic coordinates or their latent features. Then:*

1. *If $\lambda_k$ is small, the corresponding mode $\mathbf{u}_k$ represents slowly varying (global) deformations across the molecule.*

2. *If $\lambda_k$ is large, the corresponding mode $\mathbf{u}_k$ represents rapidly changing (local) structural variations.*

*Proof.* From standard results in spectral graph theory (Chung, 1997). The low-frequency (small $\lambda$) modes vary smoothly across edges, whereas high-frequency (large $\lambda$) modes exhibit large differences across edges. $\square$

### B.4 Practical Mode Truncation Criteria

**Definition B.5** (Mode Retention Threshold). For a desired tolerance $\epsilon > 0$, select $M$ such that

$$\sum_{k=M}^{N-1} \left| \mathbf{u}_k^\top \mathbf{x} \right|^2 \ \leq \ \epsilon \, \|\mathbf{x}\|_2^2. \tag{36}$$

In practice, one may also pick $M$ based on $\lambda_{M-1} \leq \alpha$, ignoring modes where $\lambda_k > \alpha$.

**Corollary B.6** (Error Control via Low-Pass Approximation). *Under the same notation as above, if*

$$\sum_{k=M}^{N-1} \left| \mathbf{u}_k^\top \mathbf{x} \right|^2 \ \leq \ \epsilon \, \|\mathbf{x}\|_2^2, \tag{37}$$

*then*

$$\|\mathbf{x} - \mathbf{x}_{(M)}\|_2 \ \leq \ \epsilon \, \|\mathbf{x}\|_2. \tag{38}$$

*Hence, discarding high-frequency modes exceeding this threshold leads to a bounded approximation error.*

## C   Temporal Dynamics of Graph Fourier Modes

We illustrate on the canonical *graph heat equation* how Laplacian eigenvalues directly determine the time-scale of each Fourier mode.

**Proposition C.1** (Heat-Equation Mode Dynamics). *Let $\mathcal{G} = (V, E)$ be a graph with normalized Laplacian $L$ admitting*

$$L = U \Lambda U^\top, \quad \Lambda = \mathrm{diag}(\lambda_0, \ldots, \lambda_{N-1}), \quad 0 = \lambda_0 \leq \lambda_1 \leq \cdots \leq \lambda_{N-1}.$$

*Consider the graph-heat evolution on a scalar signal $f(t) \in \mathbb{R}^N$:*

$$\frac{d f(t)}{dt} = -L f(t). \tag{39}$$

*Write the kth Graph Fourier coefficient as*

$$\alpha_k(t) = u_k^\top f(t).$$

*Then each mode evolves independently:*

$$\frac{d \alpha_k(t)}{dt} = u_k^\top \frac{d f(t)}{dt} = -u_k^\top L f(t) = -\lambda_k u_k^\top f(t) = -\lambda_k \alpha_k(t), \tag{40}$$

*with closed-form solution*

$$\alpha_k(t) = \alpha_k(0) \exp(-\lambda_k t).$$

*In particular:*

- *Modes with $\lambda_k$ small decay* slowly *(long time-scales).*

- *Modes with $\lambda_k$ large decay* rapidly *(short time-scales).*

*Proof.* Starting from $\frac{d f}{dt} = -L f$, project onto the orthonormal eigenvector $u_k$:

$$\frac{d \alpha_k}{dt} = u_k^\top \frac{df}{dt} = -u_k^\top L f = -(u_k^\top U) \Lambda (U^\top f) = -e_k^\top \Lambda (U^\top f) = -\lambda_k (u_k^\top f) = -\lambda_k \alpha_k.$$

The ODE $\dot{\alpha}_k = -\lambda_k \alpha_k$ integrates immediately to $\alpha_k(t) = \alpha_k(0) e^{-\lambda_k t}$. $\square$

*Remark* C.2. Although our GF-NODE dynamics are *learned* rather than the pure heat equation, Proposition 3.3 shows that under any diffusion-like operator the Laplacian eigenvalues $\lambda_k$ set intrinsic time-scales for each mode. In practice, our Neural-ODE learns a richer $f_\theta$, but we still observe empirically that modes with larger $\lambda_k$ tend to exhibit faster temporal variation—precisely the behavior we exploit by evolving each $\alpha_k(t)$ (and its vector analogue) under separate ODE channels.

### C.1   Empirical Validation of the Spatial-Temporal Scale Correspondence

Our central hypothesis is that the spatial frequencies defined by the graph Laplacian eigenvectors correspond to the characteristic timescales of molecular motion. Specifically, we claim that low-frequency spatial modes (associated with small eigenvalues $\lambda_k$) capture slow, global dynamics, while high-frequency spatial modes (large $\lambda_k$) capture fast, local vibrations. To rigorously validate this claim, we perform a joint spatial-temporal frequency analysis on the ground-truth molecular dynamics trajectories.

The procedure is as follows:

**Step 1: Obtain the Eigendecomposition of the Graph Laplacian.**   For a given molecule, we construct the graph $\mathcal{G}$ and its Laplacian $\mathbf{L} \in \mathbb{R}^{N \times N}$. We then compute its eigendecomposition:

$$\mathbf{L} = \mathbf{U} \mathbf{\Lambda} \mathbf{U}^\top$$

where $\mathbf{U} = [\mathbf{u}_0, \mathbf{u}_1, \ldots, \mathbf{u}_{N-1}]$ is the orthonormal matrix of eigenvectors, and $\mathbf{\Lambda} = \mathrm{diag}(\lambda_0, \lambda_1, \ldots, \lambda_{N-1})$ contains the corresponding real, non-negative eigenvalues sorted in ascending order ($0 = \lambda_0 \leq \lambda_1 \leq \ldots$). These eigenvectors $\{\mathbf{u}_k\}$ form the spatial frequency basis.

**Step 2: Project the Ground-Truth Trajectory onto the Spatial Basis.** Let a ground-truth trajectory be represented by a sequence of atom positions $\{\mathbf{X}(t_j)\}_{j=1}^{T_{sim}}$, where $\mathbf{X}(t_j) \in \mathbb{R}^{N \times 3}$ is the matrix of coordinates for $N$ atoms at time step $t_j$. We first ensure translational invariance by mean-centering the coordinates at each step: $\mathbf{X}_c(t_j) = \mathbf{X}(t_j) - \overline{\mathbf{X}}(t_j)$.

We then project these coordinates onto the spatial basis using the Graph Fourier Transform (GFT). For each spatial mode $k$, we obtain a time series of its 3D spectral coefficient $\tilde{\mathbf{x}}_k(t_j) \in \mathbb{R}^3$. This is computed by projecting the centered coordinates onto the eigenvector $\mathbf{u}_k$:

$$\tilde{\mathbf{x}}_k(t_j) = \mathbf{U}_{(:,k)}^\top \mathbf{X}_c(t_j) = \mathbf{u}_k^\top \mathbf{X}_c(t_j)$$

This operation yields $N$ distinct time series, $\{\tilde{\mathbf{x}}_k(t_j)\}_{j=1}^{T_{sim}}$, one for each spatial mode $k \in \{0, \ldots, N-1\}$.

**Step 3: Analyze the Temporal Frequency of Each Spatial Mode.** For each spatial mode $k$, we now have a signal $\tilde{\mathbf{x}}_k(t)$ that describes how the amplitude of that spatial pattern evolves over time. To quantify its characteristic temporal frequency, we first compute the time series of its magnitude, $s_k(t_j) = \|\tilde{\mathbf{x}}_k(t_j)\|_2$.

Next, we compute the power spectral density (PSD) of the signal $s_k(t)$ using the Discrete Fourier Transform (DFT), commonly implemented via the Fast Fourier Transform (FFT). Let the PSD be $P_k(f)$, where $f$ represents the temporal frequency.

To obtain a single characteristic temporal frequency $\bar{f}_k$ for each spatial mode $k$, we compute the power-weighted average frequency (i.e., the spectral centroid):

$$\bar{f}_k = \frac{\int_0^{f_{max}} f \cdot P_k(f)\, df}{\int_0^{f_{max}} P_k(f)\, df}$$

where the integral is performed over the range of relevant temporal frequencies up to the Nyquist limit $f_{max}$. This value $\bar{f}_k$ represents the average timescale on which the spatial mode $k$ is active.

**Step 4: Visualize the Spatial-Temporal Correlation.** Finally, we plot the characteristic temporal frequency $\bar{f}_k$ against its corresponding spatial frequency (the Laplacian eigenvalue $\lambda_k$). This creates a set of points $(\lambda_k, \bar{f}_k)$ for $k = 1, \ldots, N-1$ (we omit the $k = 0$ mode as it corresponds to the zero eigenvalue and has no dynamics). A clear positive correlation in this plot provides strong empirical evidence that spatially smoother modes (low $\lambda_k$) indeed evolve more slowly (low $\bar{f}_k$), and spatially oscillatory modes (high $\lambda_k$) evolve more quickly (high $\bar{f}_k$).

## C.2 Theoretical Framework for Joint Spatial-Temporal Analysis

To rigorously ground our model's architecture, we introduce a formal framework for analyzing the joint spatial-temporal characteristics of molecular dynamics. We treat the molecular trajectory as a time-varying signal defined on a graph and leverage tools from spectral graph theory and statistical signal processing to decompose and analyze its structure.

### C.2.1 Molecular Configuration as a Signal in a Hilbert Space

Let the molecular graph be $\mathcal{G} = (\mathcal{V}, \mathcal{E})$, with $|\mathcal{V}| = N$. The set of all possible scalar functions on the vertices, $g : \mathcal{V} \to \mathbb{R}$, forms an $N$-dimensional Hilbert space $\mathcal{H} = \ell^2(\mathcal{V})$ equipped with the standard inner product $\langle f, g \rangle = \sum_{i \in \mathcal{V}} f(i)g(i)$.

The graph Laplacian $\mathbf{L} \in \mathbb{R}^{N \times N}$ is a self-adjoint, positive semi-definite operator on this space. By the spectral theorem, it admits an orthonormal basis of eigenvectors $\{\mathbf{u}_k\}_{k=0}^{N-1}$ for $\mathcal{H}$. These eigenvectors are the natural analogues of the Fourier basis on Euclidean domains.

**Definition C.3** (Spatial Frequency Basis)**.** The eigenvectors $\{\mathbf{u}_k\}$ of the graph Laplacian $\mathbf{L}$, ordered by their corresponding eigenvalues $0 = \lambda_0 \leq \lambda_1 \leq \cdots \leq \lambda_{N-1}$, are defined as the **spatial frequency basis** of the graph $\mathcal{G}$. The eigenvalue $\lambda_k$ is interpreted as the frequency of the mode $\mathbf{u}_k$.

The frequency $\lambda_k$ quantifies the spatial variation of its mode. This is formalized by the concept of *Total Variation*. The Total Variation of a graph signal $\mathbf{f}$ is given by the Laplacian quadratic form:

$$\mathrm{TV}(\mathbf{f}) = \mathbf{f}^\top \mathbf{L} \mathbf{f} = \sum_{(i,j) \in \mathcal{E}} w_{ij} (f_i - f_j)^2$$

For an eigenmode $\mathbf{u}_k$, its total variation is simply $\mathbf{u}_k^\top \mathbf{L} \mathbf{u}_k = \lambda_k$. Thus, a small $\lambda_k$ implies the mode is spatially smooth, while a large $\lambda_k$ implies it is highly oscillatory.

### C.2.2 Spectral Representation of a Time-Varying Graph Signal

We formalize the molecular trajectory as a time-varying, vector-valued graph signal, $\mathbf{F} : \mathbb{R} \to (\ell^2(\mathcal{V}))^3$, which maps time $t$ to the matrix of atomic coordinates $\mathbf{X}(t) \in \mathbb{R}^{N \times 3}$. To analyze this signal, we project it onto the spatial frequency basis using the Graph Fourier Transform (GFT).

The GFT of the trajectory $\mathbf{F}(t)$ is a set of time-varying spectral coefficients $\{\tilde{\mathbf{x}}_k(t)\}_{k=0}^{N-1}$, where $\tilde{\mathbf{x}}_k(t) \in \mathbb{R}^3$. Each coefficient represents the projection of the molecular configuration onto a specific spatial mode at time $t$:

$$\tilde{\mathbf{x}}_k(t) = \langle \mathbf{F}(t), \mathbf{u}_k \rangle_{\mathcal{H}} := (\mathbf{u}_k^\top \mathbf{X}_{:,1}(t), \mathbf{u}_k^\top \mathbf{X}_{:,2}(t), \mathbf{u}_k^\top \mathbf{X}_{:,3}(t))^\top$$

The inverse GFT perfectly reconstructs the signal: $\mathbf{F}(t) = \sum_{k=0}^{N-1} \tilde{\mathbf{x}}_k(t) \mathbf{u}_k^\top$. From a dynamical systems perspective, the evolution of the molecule, governed by complex coupled equations in the vertex domain, can be viewed as the superposition of the dynamics of these individual modes $\tilde{\mathbf{x}}_k(t)$ in the spectral domain.

### C.2.3 Frequency Analysis of Temporal Mode Dynamics

Our core hypothesis is that the dynamics of these modes are not uniform; specifically, the characteristic temporal frequency of $\tilde{\mathbf{x}}_k(t)$ should be correlated with its spatial frequency $\lambda_k$. To quantify this, we analyze the temporal signal associated with each spatial mode.

For each mode $k$, we define a scalar time series representing its energetic contribution, $s_k(t) = \|\tilde{\mathbf{x}}_k(t)\|_2^2$. The theoretical foundation for analyzing the frequency content of such a signal is the **Wiener-Khinchin theorem**, which connects the power spectral density (PSD) of a wide-sense stationary process to the Fourier transform of its autocorrelation function. Assuming local stationarity in the dynamics, we can estimate the PSD, denoted $\mathcal{P}_k(f)$, by computing the squared magnitude of the temporal Fourier transform (FFT) of the signal $s_k(t)$.

The PSD $\mathcal{P}_k(f)$ reveals how the energy of the spatial mode $k$ is distributed across different temporal frequencies $f$. To summarize this distribution into a single characteristic frequency, we compute the **spectral centroid** $\bar{f}_k$:

$$\bar{f}_k = \frac{\int_0^\infty f \cdot \mathcal{P}_k(f) \, df}{\int_0^\infty \mathcal{P}_k(f) \, df} \tag{41}$$

The centroid $\bar{f}_k$ represents the power-weighted average frequency, providing a robust measure of the dominant timescale of the mode's dynamics.

### C.2.4 The Joint Spatial-Temporal Spectrum

By performing this analysis for each spatial mode $k \in \{1, \ldots, N-1\}$, we can construct an empirical **joint spatial-temporal spectrum** of the molecular dynamics. This spectrum is the set of points:

$$\mathcal{S} = \{(\lambda_k, \bar{f}_k)\}_{k=1}^{N-1} \subset \mathbb{R}^+ \times \mathbb{R}^+$$

### C.3 Empirical Validation Results

We validate our spatial-temporal correspondence hypothesis through comprehensive analysis of four distinct molecular systems: buckyball-catcher (MD22), salicylic acid, naphthalene, and uracil (rMD17). Following the four-step protocol outlined above, we analyze ground-truth molecular dynamics trajectories to quantify the correlation between spatial frequencies $\lambda_k$ and characteristic temporal frequencies $\bar{f}_k$.

**Experimental Setup and Data Processing.**    For each molecular system, we apply the four-step protocol described in the preceding section. The molecular systems span diverse chemical environments: buckyball-catcher represents a large supramolecular complex with 120 heavy atoms, while salicylic acid, naphthalene, and uracil are smaller organic molecules with distinct aromatic and heterocyclic structures. This diversity enables assessment of the generality of the spatial-temporal correspondence across different molecular architectures and dynamical regimes.

**Quantitative Correlation Analysis.**    We observe consistently positive correlations between spatial frequencies $\lambda_k$ and characteristic temporal frequencies $\bar{f}_k$ across all molecular systems (Figure 9). The correlation coefficients range from moderate ($r = 0.424$) to very strong ($r = 0.816$), with all systems exhibiting positive slopes in the log-log representation. Specifically, buckyball-catcher exhibits $r = 0.424$ with slope $\alpha = 0.552$, salicylic acid shows $r = 0.816$ with slope $\alpha = 2.642$, naphthalene demonstrates $r = 0.622$ with slope $\alpha = 5.016$, and uracil yields $r = 0.716$ with slope $\alpha = 4.098$. These results provide quantitative validation of the predicted power-law relationship $\bar{f}_k \propto \lambda_k^\alpha$.

**Physical Interpretation and Theoretical Validation.**    The consistently positive correlations provide empirical validation of our central hypothesis: spatial modes with larger Laplacian eigenvalues (high spatial frequency) exhibit faster characteristic temporal dynamics (high temporal frequency). This validates the theoretical prediction that the eigenspectrum of the molecular graph provides a natural ordering of dynamical timescales, with smooth, collective modes evolving slowly and oscillatory, localized modes evolving rapidly.

The variation in correlation strength and slopes across molecules reflects differences in their structural complexity and dynamical behavior. The buckyball-catcher system exhibits moderate correlation ($r = 0.424$) due to its complex multi-scale dynamics as a large supramolecular assembly. In contrast, the smaller organic molecules demonstrate stronger correlations ($r = 0.622 - 0.816$), consistent with more regular vibrational spectra and clearer timescale separation.

**Connection to Heat Equation Dynamics.**    These empirical findings provide direct validation of our theoretical derivations based on heat equation mode dynamics. In the diffusion framework, solutions of $\frac{\partial u}{\partial t} = -\mathbf{L}u$ take the form $u_k(t) = u_k(0)e^{-\lambda_k t}$, where the decay timescale $\tau_k = 1/\lambda_k$ is inversely proportional to the eigenvalue. This predicts positive correlation between spatial frequency $\lambda_k$ and temporal frequency $f_k \propto 1/\tau_k \propto \lambda_k$.

The observed positive slopes ($\alpha = 0.552 - 5.016$) confirm the power-law relationship $\bar{f}_k \propto \lambda_k^\alpha$ predicted by our theoretical analysis. The consistently positive slopes validate that the graph Laplacian eigenspectrum provides a natural frequency ordering for molecular dynamics, bridging spatial structure and temporal evolution through spectral graph theory.

**Implications for Physics-Informed Neural Architectures.**    The demonstrated spatial-temporal correspondence validates graph Laplacian eigenmodes as a physics-informed basis for neural network architectures in molecular dynamics. By aligning representational capacity with the natural frequency hierarchy, Fourier-based models can achieve more efficient and physically meaningful learning. The positive correlations further suggest that truncated representations retaining low-frequency spatial modes will preferentially capture slow, collective motions dominating long-timescale behavior, providing theoretical justification for dimensionality reduction strategies in molecular machine learning.

# D   Formal Proof of $SO(3)$-Equivariance for GF-NODE Pipeline

Below is a **formal proof** of SO(3) (rotational) equivariance for our **GF-NODE** pipeline, closely following the style of EGNO's Appendix proofs. We focus on the **3D rotational** part of SE(3); translations can be handled by the separate mean-centering step (see remarks below). Our proof is broken down into:

1. **Defining the R-action**,
2. **Showing that each module** (Fourier transforms, block-diagonal ODE, EGNN layers) **is** SO(3)-**equivariant**, and
3. **Composing these results** to conclude overall equivariance.

## D.1   Formal Statement of $SO(3)$-Equivariance

Let

$$f = \begin{bmatrix} f_{\mathbf{h}}, f_{\mathbf{Z}} \end{bmatrix}^{\top} \tag{42}$$

be a function describing the node features of a 3D molecular system over some (possibly temporal) domain $D$. Concretely,

- $f_{\mathbf{h}} : D \to \mathbb{R}^{N \times k}$ collects **invariant (scalar) node features**,
- $f_{\mathbf{Z}} : D \to \mathbb{R}^{N \times (m \times 3)}$ collects **equivariant (3D) features** (positions, velocities, etc.).

Denote by $\mathbf{R} \in SO(3)$ a 3D rotation matrix. The **action** of $\mathbf{R}$ on $f$ is defined by

$$(\mathbf{R} \cdot f)(t) = \begin{bmatrix} f_{\mathbf{h}}(t), \mathbf{R} f_{\mathbf{Z}}(t) \end{bmatrix}^{\top}, \tag{43}$$

which rotates only the $\mathbf{Z}$-component in $\mathbb{R}^3$ and leaves the scalar $\mathbf{h}$-component invariant.

We claim that our overall **GF-NODE operator** $\mathcal{T}_{\theta}$ satisfies

$$\mathbf{R} \cdot \mathcal{T}_{\theta}(f) = \mathcal{T}_{\theta}(\mathbf{R} \cdot f), \tag{44}$$

i.e., $\mathcal{T}_{\theta}$ is SO(3)-equivariant. Formally:

**Theorem D.1** (SO(3) Equivariance). *Let $\mathcal{T}_{\theta}$ be the GF-NODE architecture composed of:*

*1. An **EGNN encoder** (mapping $[f_{\mathbf{h}}, f_{\mathbf{Z}}] \to$ encoded features),*
*2. **Mean-centering** and **Graph Fourier Transform** ($\mathcal{F}$),*
*3. A **block-diagonal Neural ODE** in the spectral domain,*
*4. **Inverse GFT** ($\mathcal{F}^{-1}$) plus adding back the mean, and*
*5. An **EGNN decoder**.*

*Then for any $\mathbf{R} \in SO(3)$, the pipeline satisfies*

$$\mathcal{T}_{\theta}(\mathbf{R} \cdot f) = \mathbf{R} \cdot \mathcal{T}_{\theta}(f). \tag{45}$$

*In other words, rotating the input 3D features by $\mathbf{R}$ is equivalent to applying $\mathcal{T}_{\theta}$ first and then rotating the result.*

We prove this via the following steps:

1. **Lemma 1**: EGNN layers are SO(3)-equivariant.
2. **Lemma 2**: GFT and its inverse are SO(3)-equivariant (dimension-wise linearity).
3. **Lemma 3**: The block-diagonal Neural ODE in spectral space preserves $\mathbf{R}$-equivariance on the vector channels.
4. **Conclusion**: Composing these yields the full pipeline's equivariance.

Below, we provide the details of each lemma and then the final proof of the Theorem.

## D.2 EGNN Equivariance

**Lemma D.2** (EGNN layers are SO(3)-equivariant). *Consider a generic EGNN layer $\Phi$, which updates*

$$(\mathbf{h}_i, \mathbf{x}_i) \mapsto (\mathbf{h}'_i, \mathbf{x}'_i), \tag{46}$$

*using message passing:*

$$\mathbf{m}_{ij} = \phi_e\big(\mathbf{h}_i, \mathbf{h}_j, \mathbf{x}_i - \mathbf{x}_j\big), \tag{47}$$

$$\mathbf{h}'_i = \phi_h\Big(\mathbf{h}_i, \sum_j \mathbf{m}_{ij}\Big), \tag{48}$$

$$\mathbf{x}'_i = \mathbf{x}_i + \ldots (\mathbf{x}_i - \mathbf{x}_j). \tag{49}$$

*Then for any rotation $\mathbf{R} \in \mathrm{SO}(3)$,*

$$\Phi\big(\mathbf{R}\,\mathbf{x}_i, \mathbf{h}_i\big) = \big(\mathbf{h}'_i, \mathbf{R}\,\mathbf{x}'_i\big). \tag{50}$$

*Hence $\Phi$ is $\mathrm{SO}(3)$-equivariant on its 3D inputs.*

A standard proof ( ) shows that each update depends on $\mathbf{x}_i - \mathbf{x}_j$, which under a global rotation $\mathbf{R}(\mathbf{x}_i - \mathbf{x}_j)$ transforms consistently to yield $\mathbf{R}\,\mathbf{x}'_i$. The same argument applies to 3D velocities (or any additional 3D vectors).

## D.3 GFT Equivariance

We next show that the (inverse) Graph Fourier Transform is SO(3)-equivariant with respect to dimension-wise rotations of the 3D features.

**Lemma D.3** (GFT and $\mathcal{F}^{-1}$ are $\mathrm{SO}(3)$-equivariant)). *Let $\mathcal{F}$ be the dimension-wise GFT mapping a function*

$$f_{\mathbf{Z}} : D \to \mathbb{R}^{N \times (m \times 3)} \tag{51}$$

*to its frequency coefficients $\mathcal{F}(f_{\mathbf{Z}}) \in \mathbb{C}^{(modes) \times m \times 3}$. Under $\mathbf{R} \in \mathrm{SO}(3)$, define*

$$\mathbf{R} \cdot \big(\mathcal{F}f_{\mathbf{Z}}\big) = \mathcal{F}f_{\mathbf{Z}} \text{ but with each } 3\text{-}D \text{ channel rotated by } \mathbf{R}. \tag{52}$$

*Then*

$$\mathbf{R} \cdot \mathcal{F}(f_{\mathbf{Z}}) = \mathcal{F}\Big(\mathbf{R} \cdot f_{\mathbf{Z}}\Big). \tag{53}$$

*Similarly, $\mathcal{F}^{-1}$ is $\mathrm{SO}(3)$-equivariant in the sense that*

$$\mathcal{F}^{-1}\big(\mathbf{R} \cdot F\big) = \mathbf{R} \cdot \mathcal{F}^{-1}(F). \tag{54}$$

**Proof Sketch.** The GFT (and its inverse) act linearly along each 3D axis. If $\mathbf{R}$ rotates the 3D channels, we can commute $\mathbf{R}$ with the linear transform $\mathcal{F}$. Precisely as in the EGNO proof, the multilinear expansions show that $\mathbf{R} \cdot \mathcal{F}(f_{\mathbf{Z}}) = \mathcal{F}\big(\mathbf{R} \cdot f_{\mathbf{Z}}\big)$. The same argument applies to $\mathcal{F}^{-1}$ because it is also linear and dimension-wise.

## D.4 Block-Diagonal Neural ODE Equivariance

In the GF-NODE pipeline, once we have GFT coefficients $\tilde{\mathbf{Z}}$, the **Neural ODE** acts as a block-diagonal operator:

$$\begin{pmatrix} \tilde{\mathbf{H}} \\ \tilde{\mathbf{Z}} \end{pmatrix} \mapsto \begin{pmatrix} f_\theta\big(\tilde{\mathbf{H}}\big) \\ g_\theta\big(\tilde{\mathbf{Z}}\big) \end{pmatrix}, \tag{55}$$

where $\tilde{\mathbf{Z}} \in \mathbb{C}^{(\mathrm{modes}) \times m \times 3}$. Rotating $\mathbf{R}$ on these 3D channels amounts to mixing the coordinate axes linearly. Because the ODE is chosen to be channelwise or "blockwise" linear or MLP-based, it commutes with $\mathbf{R}$. Hence:

**Lemma D.4** (Block-Diagonal ODE is $\mathrm{SO}(3)$-equivariant). *For each frequency mode, the update on $\tilde{\mathbf{Z}}$ is dimension-wise (like a separate channel). A global rotation $\mathbf{R}$ that mixes $\tilde{\mathbf{Z}}^1, \tilde{\mathbf{Z}}^2, \tilde{\mathbf{Z}}^3$ can be factored out of the ODE solution—so*

$$\mathbf{R} \cdot g_\theta(\tilde{\mathbf{Z}}) = g_\theta\big(\mathbf{R} \cdot \tilde{\mathbf{Z}}\big). \tag{56}$$

*Integrating over $t$ preserves this property.*

## D.5 Proof of the Main Theorem (SO(3) Equivariance)

Recall our overall operator $\mathcal{T}_\theta$ has the form:

1. **EGNN Encode**: $\big(\mathbf{h}, \mathbf{Z}\big) \mapsto \big(\mathbf{h}^{(L)}, \mathbf{Z}^{(L)}\big)$.
2. **Mean-Center + GFT**: $\mathbf{Z}^{(L)} \mapsto \mathbf{Z}_c^{(L)} \mapsto \tilde{\mathbf{Z}}^{(L)} = \mathcal{F}\big(\mathbf{Z}_c^{(L)}\big)$.
3. **Block-Diagonal ODE**: $\tilde{\mathbf{Z}}^{(L)} \mapsto \tilde{\mathbf{Z}}(t)$ for any $t$.
4. **Inverse GFT + Add Mean**: $\tilde{\mathbf{Z}}(t) \mapsto \mathbf{Z}_c(t) = \mathcal{F}^{-1}\big(\tilde{\mathbf{Z}}(t)\big) \mapsto \mathbf{Z}(t)$.
5. **EGNN Decode**: $\big(\mathbf{h}, \mathbf{Z}(t)\big) \mapsto \big(\mathbf{h}'(t), \mathbf{Z}'(t)\big)$.

To show $\mathbf{R} \cdot \mathcal{T}_\theta(f) = \mathcal{T}_\theta(\mathbf{R} \cdot f)$, we proceed step-by-step:

1. **EGNN Encode**: By Lemma A.1, if the input positions are replaced with $\mathbf{R}\mathbf{x}_i$, the output is $\mathbf{R}\mathbf{x}_i^{(L)}$.
2. **Mean-Center**: Under a global rotation, the centered coordinates also rotate, i.e., $\mathbf{x}_i^\circ \mapsto \mathbf{R}\,\mathbf{x}_i^\circ$.
3. **GFT**: By Lemma A.2, dimension-wise GFT on $\mathbf{R}\,\mathbf{x}_i^\circ$ yields the rotated spectral coefficients.
4. **Block-Diagonal ODE**: Lemma A.3 says the ODE in spectral space is equivariant w.r.t. 3D axis mixing, so $\mathbf{R}$ commutes with the ODE solution.
5. **Inverse GFT**: Again by Lemma A.2, inverse transforms are linear in each dimension, preserving $\mathbf{R}$ on the output.
6. **Add Mean**: The final global shift (if any) is consistent with $\mathbf{R}$.
7. **EGNN Decode**: By Lemma A.1 again, if the input to the decoder is rotated, the output is the rotated version of the unrotated output.

Hence each sub-module respects the action of $\mathbf{R}$. Composing them in order yields the final statement

$$\mathcal{T}_\theta(\mathbf{R} \cdot f) = \mathbf{R} \cdot \mathcal{T}_\theta(f). \tag{57}$$

This completes the proof of SO(3)-equivariance.

## D.6 Remarks on Translations

In practice, **SE(3)** includes translations as well. Our pipeline **removes** the translational degree of freedom by **mean-centering** the positions (the DC mode). A global translation $\mathbf{x}_i \mapsto \mathbf{x}_i + \boldsymbol{\mu}$ simply shifts the mean $\bar{\mathbf{x}}$, so the centered coordinates $\mathbf{x}_i^\circ$ remain unchanged. This effectively "factors out" translation before the GFT steps. When we **re-add** the mean at the end, it ensures the final positions transform by $\mathbf{x}_i \mapsto \mathbf{x}_i + \boldsymbol{\mu}$. Thus the entire pipeline remains **invariant** to translations (i.e., translates its outputs accordingly). For brevity, the above proof focuses on **rotations $\mathbf{R} \in \mathrm{SO}(3)$**; translation invariance follows from the mean-subtraction procedure plus the decoder's reliance on relative positions.

# E   Additional Experiment Results

Below, we provide four tables corresponding to different experimental comparisons. The first table reports performance on the MD17 dataset with regular (equi-) timesteps. The remaining three tables focus on ablation experiments conducted under irregular sampling conditions: (i) ablations on the alanine dipeptide dataset, (ii) ablations comparing different GNN architectures, and (iii) ablations on different temporal embedding approaches.

Table 7: MD17 with Regular (Equi-) Timestep Sampling. MSE ($\times 10^{-2}$ Å$^2$) on the MD17 dataset using regular timesteps. Best results are in **bold**, and second-best are underlined.

|  | Aspirin | Benzene | Ethanol | Malonaldehyde | Naphthalene | Salicylic | Toluene | Uracil |
|---|---|---|---|---|---|---|---|---|
| NDCN | 31.73±0.40 | 56.21±0.30 | 10.74±0.02 | 46.55±0.28 | 2.25±0.01 | 3.58±0.11 | 13.92±0.02 | 2.38±0.00 |
| LG-ODE | 19.36±0.12 | 53.92±1.32 | 7.08±0.01 | 24.41±0.03 | 1.73±0.02 | 3.82±0.04 | 11.18±0.01 | 2.11±0.02 |
| EGNN | 9.24±0.07 | 57.85±2.70 | 4.63±0.00 | 12.81±0.01 | 0.38±0.01 | 0.85±0.00 | 10.41±0.04 | 0.56±0.02 |
| EGNO | 9.41±0.09 | 55.13±3.21 | 4.63±0.00 | 12.81±0.01 | 0.40±0.01 | 0.93±0.01 | 10.43±0.10 | 0.59±0.01 |
| ITO | 20.56±0.03 | 57.25±0.58 | 8.60±0.27 | 28.44±0.73 | 1.82±0.17 | 2.48±0.34 | 12.47±0.30 | 1.33±0.12 |
| Ours | **6.07**±0.09 | **1.51**±0.07 | **2.74**±0.01 | **9.43**±0.02 | **0.24**±0.02 | **0.63**±0.05 | **1.80**±0.03 | **0.41**±0.02 |

**Explanation.**   Table 7 shows results for MD17 under a regular (evenly spaced) sampling scheme. Although the dataset inherently has fine-grained timesteps, we constrain both training and evaluation to equidistant frames to compare methods fairly. Our approach demonstrates consistent improvements over baselines on nearly all molecules.

Table 8: **Ablation Results on Alanine Dipeptide (Irregular Sampling).** MSE ($\times 10^{-3}$ nm$^2$). Best results in **bold**.

|  | standard | no_ode | no_ode_h | no_ode_x | no_interaction |
|---|---|---|---|---|---|
| MSE | **4.48**±0.07 | 4.72±0.05 | 4.60±0.05 | 4.64±0.05 | 4.80±0.04 |

|  | interaction_concat | time_posenc | time_mlp | FFT | no_fourier |
|---|---|---|---|---|---|
| MSE | 4.98±0.06 | 4.62±0.05 | 4.60±0.05 | 4.57±0.05 | 4.51±0.04 |

**Explanation.**   Table 8 organizes the ablation settings into two rows, each containing five columns. The first row compares our "standard" model to variants that remove specific ODE blocks or modify scalar/vector-only ODE updates ("no_ode","no_ode_h", "no_ode_x"), and the second row compares different interaction modes, time embeddings, and Fourier settings. The "standard" configuration achieves the best overall MSE.

Table 9: **Ablation on GNN Architectures (Irregular Sampling).** MSE ($\times 10^{-2}$ Å$^2$) on MD17 comparing different GNN layers (SAGEConv, GCNConv, EGNNConv). Best results in **bold**.

|  | Aspirin | Benzene | Ethanol | Malonaldehyde | Naphthalene | Salicylic | Toluene | Uracil |
|---|---|---|---|---|---|---|---|---|
| **SAGEConv** | **6.46**±0.03 | 1.79±0.08 | **2.74**±0.05 | **10.54**±0.01 | **0.23**±0.02 | **0.63**±0.01 | 3.08±0.05 | **0.41**±0.01 |
| **GCNConv** | 6.91±0.02 | **1.52**±0.08 | 3.09±0.06 | 10.85±0.03 | 0.42±0.01 | 0.88±0.00 | **1.80**±0.05 | 0.61±0.02 |
| **EGNNConv** | 8.85±0.02 | 40.86±0.98 | 4.41±0.06 | 12.49±0.00 | 0.40±0.01 | 0.87±0.01 | 8.63±0.04 | 0.62±0.02 |

**Explanation.**   Table 9 shows how our model performs with different GNN backbones on MD17 under irregular sampling. Overall, SAGEConv yields robust performance for most molecules, whereas GCNConv provides better results specifically on Benzene and Toluene. EGNNConv performs well on some local metrics but struggles on large translations (i.e., Benzene).

**Explanation.**   Table 10 compares three different time-embedding methods under irregular timestep sampling: positional encoding (posenc), a small MLP (mlp), and a direct concatenation of time tokens (concat).

Table 10: **Ablation on Time Embedding Approaches (Irregular Sampling).** MSE ($\times 10^{-2}$ Å$^2$) on MD17 across different time encoding schemes.

|  | Aspirin | Benzene | Ethanol | Malonaldehyde | Naphthalene | Salicylic | Toluene | Uracil |
|---|---|---|---|---|---|---|---|---|
| posenc | 6.91±0.08 | 1.81±0.02 | 3.11±0.05 | 10.69±0.03 | 4.19±0.04 | 0.87±0.01 | 3.56±0.07 | 0.59±0.02 |
| mlp | 6.61±0.06 | 1.61±0.03 | 3.08±0.02 | 10.62±0.04 | 0.41±0.01 | 0.87±0.02 | 3.25±0.06 | 0.56±0.01 |
| concat | **6.46**±0.03 | **1.52**±0.08 | **2.74**±0.05 | **10.54**±0.01 | **0.23**±0.02 | **0.63**±0.01 | **1.80**±0.05 | **0.41**±0.01 |

Concatenation achieves the lowest MSE, suggesting that a straightforward inclusion of time in the feature vector can be beneficial, though the MLP variant also achieves competitive performance on several molecules.

Table 11: Comparison of GF-NODE with baseline models on the revised MD17 dataset at $\Delta t = 3000$. MSE ($\times 10^{-2}$ Å$^2$) values; best results in **bold**.

| Model | Aspirin | Azobenzene | Ethanol | Malonaldehyde | Naphthalene | Paracetamol | Salicylic | Toluene | Uracil |
|---|---|---|---|---|---|---|---|---|---|
| NDCN | 34.78±0.57 | 8.45±0.29 | 24.67±0.22 | 39.02±0.51 | 1.28±0.04 | 27.13±0.41 | 1.08±0.03 | 25.99±0.36 | 0.88±0.05 |
| LG-ODE | 33.40±0.15 | 9.88±0.34 | 23.15±0.17 | 41.21±0.64 | 1.42±0.06 | 26.17±0.22 | 1.33±0.05 | 24.75±0.27 | 0.95±0.03 |
| EGNN | 31.45±0.29 | 11.03±0.41 | 22.95±0.19 | 38.80±0.30 | 1.18±0.07 | 25.87±0.30 | 1.20±0.04 | 23.90±0.19 | 0.82±0.02 |
| EGNO | 32.01±0.83 | 7.51±0.12 | 23.58±0.39 | 37.90±0.47 | 1.37±0.05 | 26.02±0.36 | 0.88±0.02 | 24.82±0.65 | 0.78±0.04 |
| ITO | 38.50±1.02 | 10.87±0.53 | 25.33±0.71 | 43.55±0.92 | 1.69±0.09 | 28.45±0.28 | 1.66±0.07 | 27.35±0.59 | 1.12±0.11 |
| GF-NODE | **30.27**±0.04 | **7.03**±0.02 | **21.92**±0.03 | **37.92**±0.05 | **1.10**±0.01 | **24.46**±0.04 | **0.81**±0.01 | **23.13**±0.04 | **0.62**±0.01 |

Table 12: Comparison of GF-NODE with baseline models on the revised MD17 dataset at $\Delta t = 10000$. MSE ($\times 10^{-2}$ Å$^2$) values; best results in **bold**.

| Model | Aspirin | Azobenzene | Ethanol | Malonaldehyde | Naphthalene | Paracetamol | Salicylic | Toluene | Uracil |
|---|---|---|---|---|---|---|---|---|---|
| NDCN | 42.67±0.91 | 11.34±0.72 | 29.45±0.47 | 48.75±1.10 | 1.90±0.03 | 33.83±0.59 | 1.95±0.22 | 34.12±0.48 | 1.72±0.09 |
| LG-ODE | 46.12±0.37 | 9.88±0.27 | 31.05±0.33 | 44.80±0.68 | 2.13±0.07 | 29.05±0.31 | 1.65±0.26 | 30.48±0.19 | 1.22±0.04 |
| EGNN | 38.09±0.16 | 13.67±0.41 | 26.14±0.26 | 41.95±0.21 | 2.07±0.13 | 28.45±0.17 | 1.27±0.08 | 29.83±0.28 | 1.01±0.05 |
| EGNO | 40.99±0.54 | 12.39±0.22 | 27.88±0.39 | 42.33±0.94 | 2.22±0.04 | 27.12±0.47 | 1.58±0.10 | 32.15±0.26 | 0.95±0.02 |
| ITO | 49.77±1.12 | 15.03±0.67 | 34.11±0.82 | 53.50±0.73 | 2.56±0.14 | 35.98±0.65 | 2.22±0.17 | 36.45±0.54 | 1.83±0.07 |
| GF-NODE | **33.18**±0.03 | **7.29**±0.03 | **22.31**±0.04 | **38.74**±0.05 | **1.27**±0.01 | **27.20**±0.04 | **0.93**±0.01 | **27.92**±0.04 | **0.72**±0.01 |

Table 13: MSE ($\times 10^{-2}$ Å$^2$) for Ala2 and larger molecules at $\Delta t = 3000$. Best results in **bold**, second best underlined.

| Model | Ala2 | Ac-Ala3-NHMe | AT-AT-CG-CG | Bucky-Catcher | DW Nanotube |
|---|---|---|---|---|---|
| NDCN | 122.65±1.87 | 22.34±0.22 | 26.78±0.50 | 6.10±0.15 | 4.50±0.20 |
| LG-ODE | 90.15±0.90 | 30.12±1.00 | 33.50±1.10 | 8.25±0.40 | 5.80±0.30 |
| EGNN | 56.70±0.84 | 18.45±0.12 | 20.75±0.45 | 7.10±0.25 | 5.60±0.35 |
| EGNO | 69.17±2.58 | 23.10±0.35 | 17.20±0.20 | 5.30±0.10 | 4.50±0.15 |
| ITO | 269.45±1.87 | 28.90±0.95 | 32.00±1.25 | 8.60±0.50 | 3.80±0.08 |
| GF-NODE | **44.82**±0.71 | **13.19**±0.13 | **14.07**±0.23 | **3.09**±0.04 | **2.58**±0.02 |

Table 14: MSE ($\times 10^{-2}$ Å$^2$) for Ala2 and larger molecules at $\Delta t = 10000$. Best results in **bold**, second best underlined.

| Model | Ala2 | Ac-Ala3-NHMe | AT-AT-CG-CG | Bucky-Catcher | DW Nanotube |
|---|---|---|---|---|---|
| NDCN | $134.10_{\pm 0.48}$ | $30.15_{\pm 0.26}$ | $38.82_{\pm 0.60}$ | $7.32_{\pm 0.18}$ | $5.85_{\pm 0.24}$ |
| LG-ODE | $117.20_{\pm 1.08}$ | $40.66_{\pm 1.20}$ | $48.58_{\pm 1.32}$ | $9.90_{\pm 0.48}$ | $7.54_{\pm 0.36}$ |
| EGNN | $88.63_{\pm 0.36}$ | $24.91_{\pm 0.14}$ | $30.09_{\pm 0.54}$ | $8.52_{\pm 0.30}$ | $7.28_{\pm 0.42}$ |
| EGNO | $\underline{73.71}_{\pm 1.10}$ | $31.19_{\pm 0.42}$ | $\underline{24.94}_{\pm 0.24}$ | $\underline{6.36}_{\pm 0.12}$ | $5.85_{\pm 0.18}$ |
| ITO | $297.21_{\pm 1.38}$ | $39.02_{\pm 1.14}$ | $46.40_{\pm 1.50}$ | $10.32_{\pm 0.60}$ | $\underline{4.94}_{\pm 0.10}$ |
| GF-NODE | $\mathbf{49.20}_{\pm 0.31}$ | $\mathbf{16.72}_{\pm 0.14}$ | $\mathbf{17.89}_{\pm 0.29}$ | $\mathbf{4.37}_{\pm 0.03}$ | $\mathbf{3.22}_{\pm 0.05}$ |

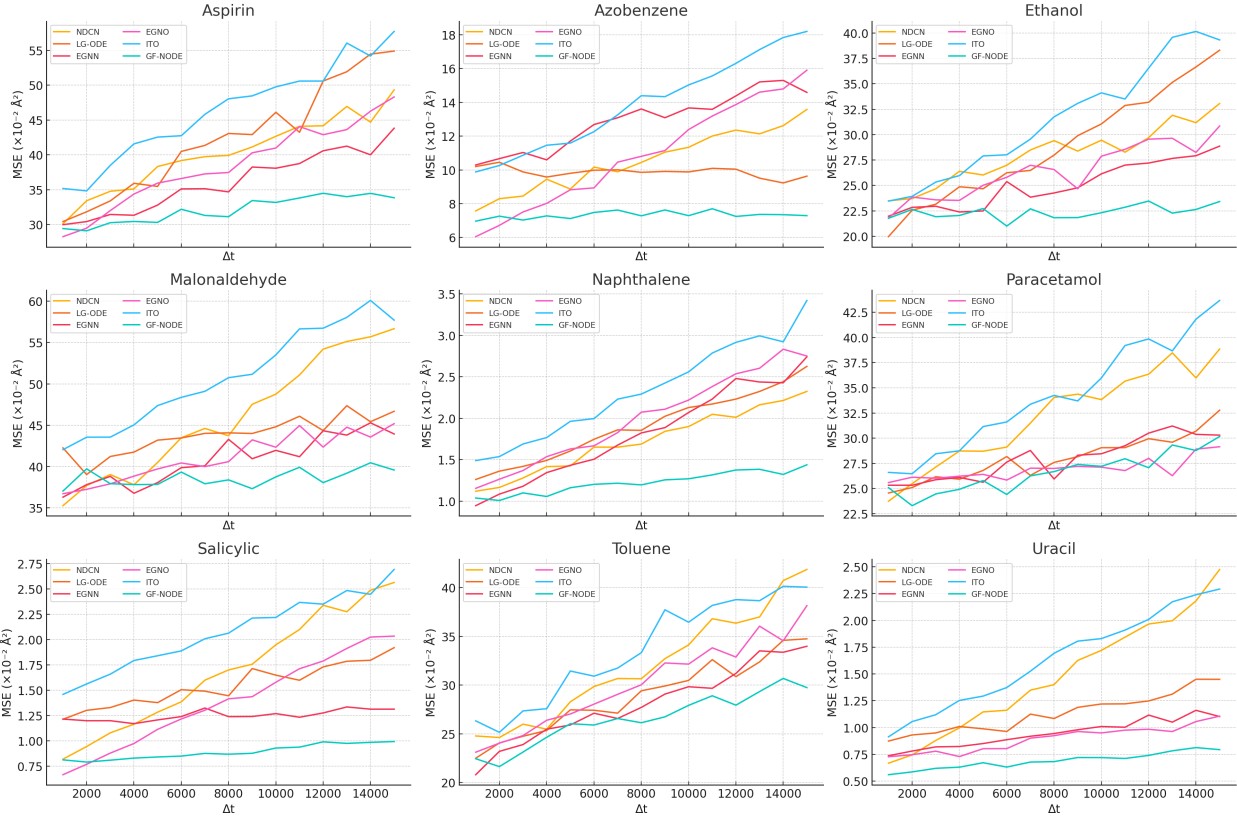

Figure 10: Temporal error growth for GF-NODE and baseline models on nine molecules on the revised MD17 dataset. Each panel plots MSE ($\times 10^{-2}$ Å$^2$) versus integration horizon $\Delta t = 1000, 2000, ..., 15000$.

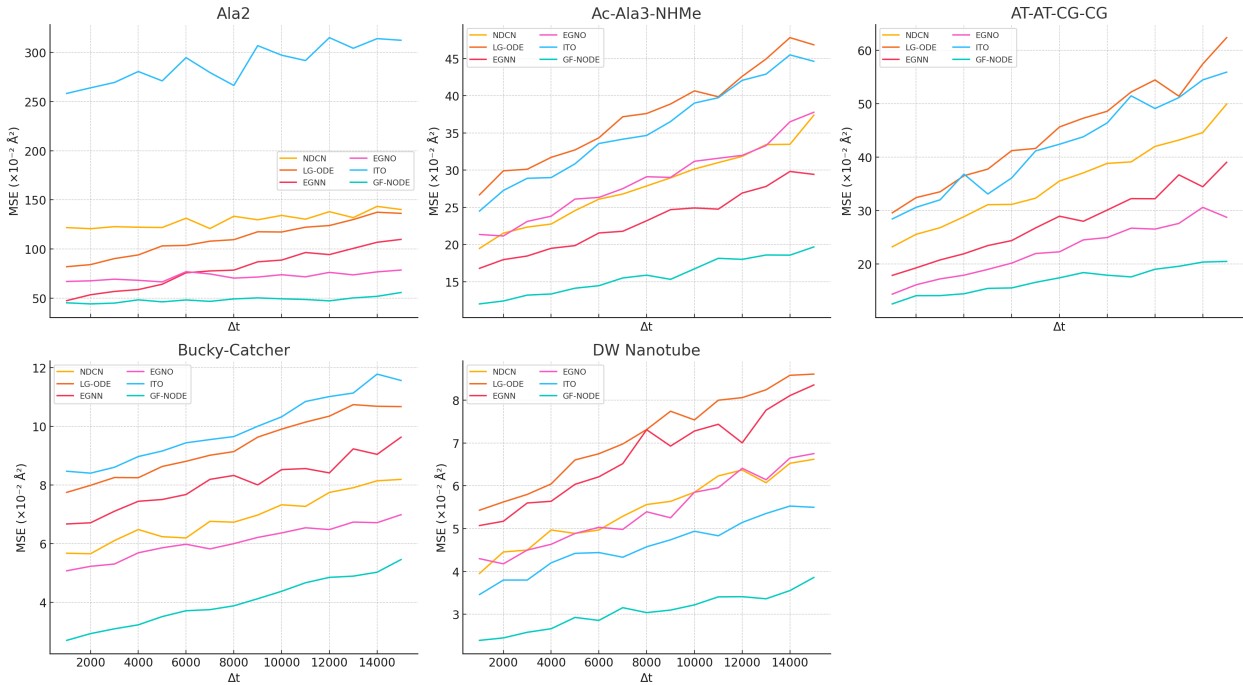

Figure 11: Long-horizon MSE trends for five larger molecules (Ala$_2$, Ac-Ala$_3$-NHMe, AT-AT-CG-CG, Bucky-Catcher, DW Nanotube). Each panel shows MSE ($\times 10^{-2}$ Å$^2$) for GF-NODE and five baselines over integration horizons $\Delta t = 1000, 2000, ..., 15000$.

Table 15: Number of heavy (non-H) atoms in each molecule.

| Molecule | Dataset | # Heavy Atoms |
|---|---|---|
| **Revised MD17** | | |
| Aspirin | | 13 |
| Azobenzene | | 14 |
| Ethanol | | 3 |
| Malonaldehyde | | 5 |
| Naphthalene | | 10 |
| Paracetamol | | 11 |
| Salicylic acid | | 10 |
| Toluene | | 7 |
| Uracil | | 8 |
| **Larger Molecular Systems** | | |
| Ala$_2$ | | 22 |
| Ac-Ala$_3$-NHMe | | 20 |
| Bucky-Catcher | | 120 |
| AT-AT-CG-CG | | 76 |
| DW Nanotube | | 326 |

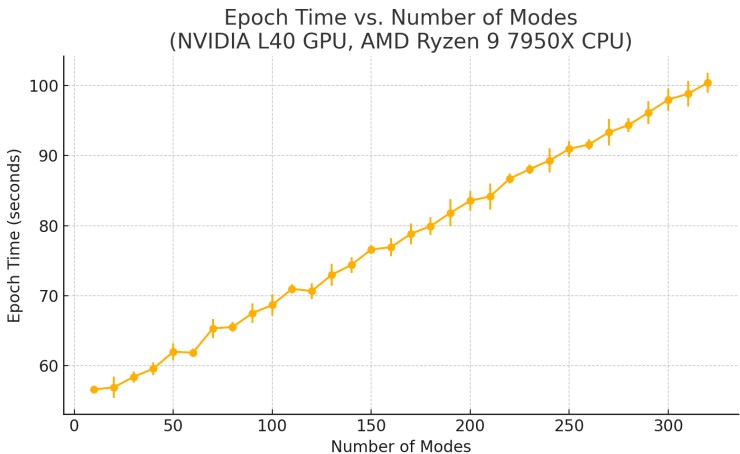

Figure 12: Epoch time (seconds) as a function of the number of Fourier modes used, measured on an NVIDIA L40 GPU with an AMD Ryzen 9 7950X CPU. Error bars represent variability across three repeated timing runs at each mode count.

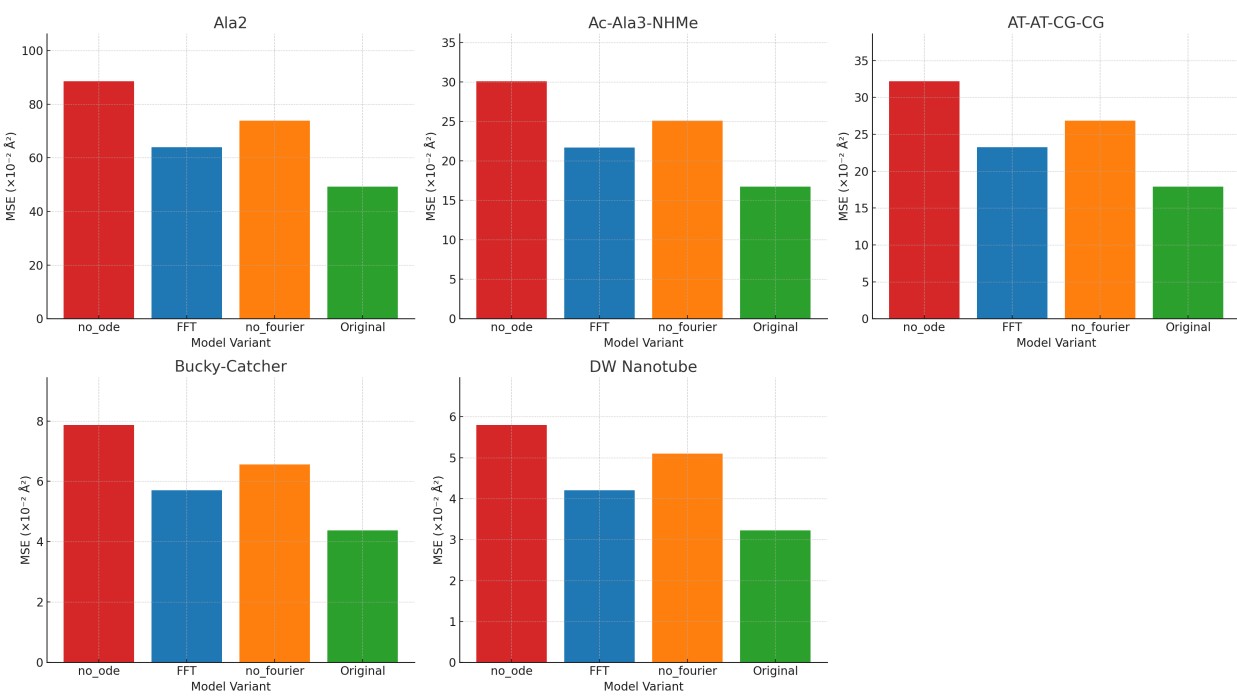

Figure 13: Ablation study on hierarchical components for five larger molecules (Ala$_2$, Ac-Ala$_3$-NHMe, AT-AT-CG-CG, Bucky-Catcher, DW Nanotube) at $\Delta t = 10000$. Variants shown are: no_ode (red), FFT only (blue), no_fourier (orange), and the full model (Original, green). Removing ODE or Fourier components degrades performance—often exceeding baseline errors—whereas the complete architecture attains the lowest MSE.

# F   Radial Distribution Function Analysis

**Section summary.** This appendix contrasts the RDFs predicted by GF–NODE with reference *ab initio* data, both system–averaged and element–specific, to evaluate structural fidelity.

Radial distribution functions (RDF, $g(r)$) quantify how atomic density varies as a function of distance and therefore provide a stringent test of whether a learned model reproduces the local and intermediate-range structure of condensed–phase systems. Below we compare the RDFs produced by GF–NODE with those obtained from reference *ab initio* trajectories (blue).

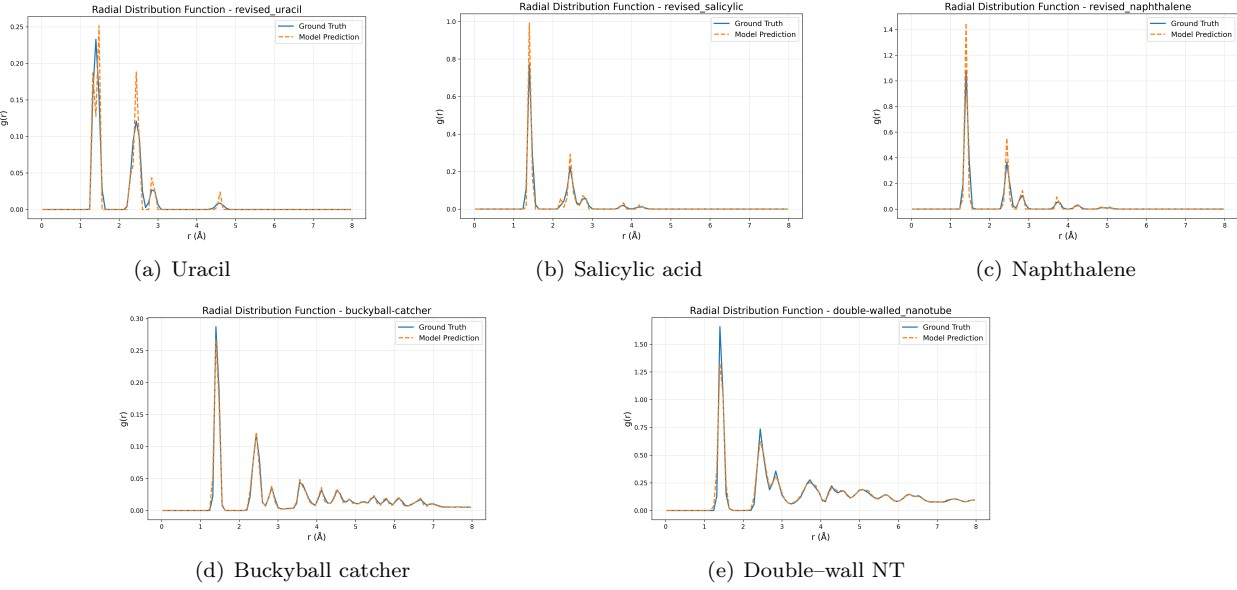

Figure 14: System–averaged radial distribution functions $g(r)$ for the five benchmark molecules/complexes. Orange: GF–NODE; black: *ab initio*. The close match indicates that the model accurately reproduces both short- and medium-range order.

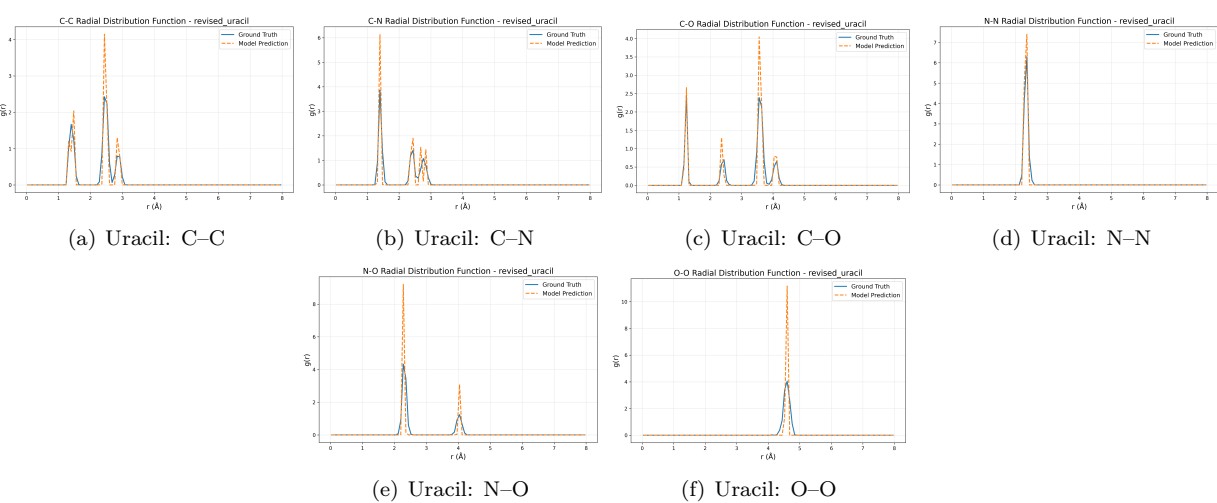

Figure 15: Element–specific RDFs $g_{\alpha\beta}(r)$ for uracil showing six unique heavy-atom pairs. GF–NODE reproduces both the peak positions and intensities of the reference curves.

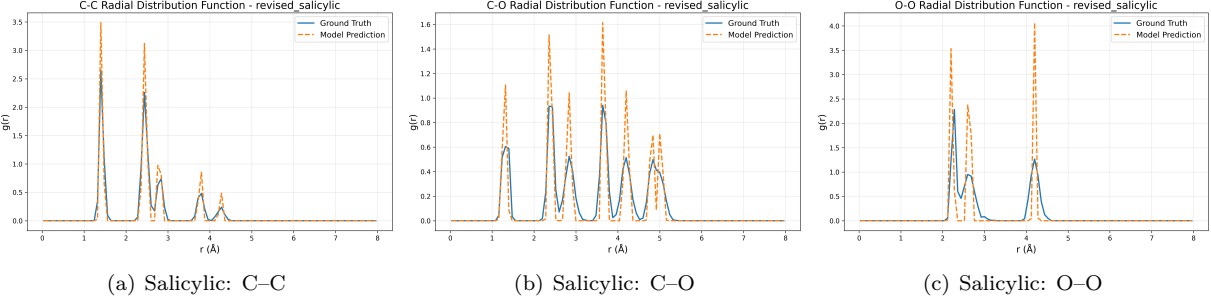

(a) Salicylic: C–C       (b) Salicylic: C–O       (c) Salicylic: O–O

Figure 16: Element–specific RDFs for salicylic acid.

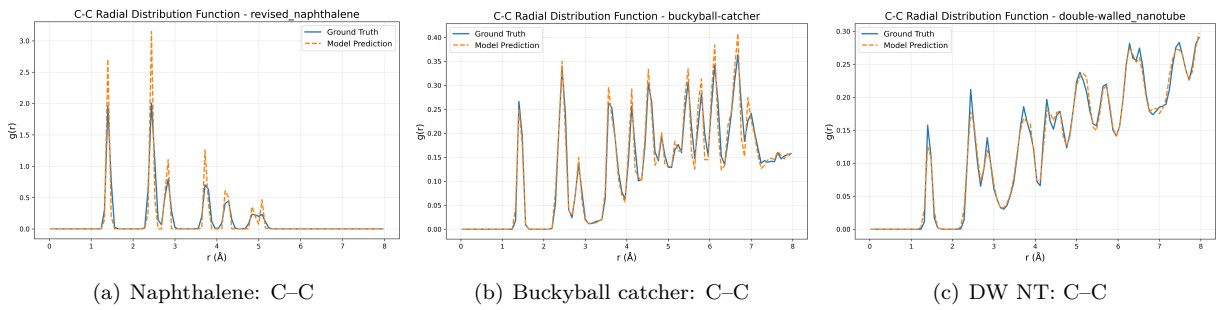

(a) Naphthalene: C–C       (b) Buckyball catcher: C–C       (c) DW NT: C–C

Figure 17: Carbon–carbon RDFs for three purely carbonaceous systems. GF–NODE captures the first–shell peak ($\approx$1.4 Å) and the longer–range oscillations characteristic of aromatic stacking and nanotube wall spacing.

