# OpenReview forum: "Graph Fourier Neural ODEs: Modeling Spatial-temporal Multi-scales in Molecular Dynamics"
_TMLR — Accepted by TMLR_

### Review · Reviewer_BeZR · 2025-04-19

**Summary Of Contributions:**

The author propose a graph neural ODE for molecular dynamics simulation. The idea is simple, first maps the node features (both scalar and vectorial) to the spectral domain. Then fit a neural ODE on the spectral domain. Finally, the node features are mapped back via inverse Fourier transform. The author verified their methods on MD17 and Alanine Dipeptide with irregular time-series sampling.

**Audience:**

Yes

**Claims And Evidence:**

Yes

**Requested Changes:**

1. **Include Loss Function Equation**:
   Explicitly present the mathematical formulation of the loss function used during model training to clarify the optimization objective.

2. **Add Proof of Equivariance**:
   Introduce a dedicated section that formally proves the equivariance properties of the proposed model, including the group actions involved and their preservation under the model’s operations.

3. **Compare with MLFF Methods**:
   Add a section discussing the relationship between the proposed method and existing MLFF approaches. Highlight some theoretical distinctions.

4. **Extend to MD22 Benchmark**:
   Demonstrate the applicability of the proposed model by evaluating its performance on the MD22 dataset.

5. **Velocity Autocorrelation Function (VACF) Analysis**:
   Plot and compare the velocity autocorrelation functions across MLFF-based methods, the proposed approach, and DFT simulations to assess dynamical consistency.

6. **Additional MD Metrics Beyond MAE**:
   Complement MAE with other molecular dynamics metrics such as the Radial Distribution Function (RDF)-based stability.

7. **Discussion on Energy Conservation**:
   Address whether the proposed method conserves energy during MD simulations. If energy is not strictly conserved, include a discussion explaining why and its implications for stability and physical realism.

**Strengths And Weaknesses:**

**Strengths:**

1. The integration of spectral methods with neural ODEs is a novel and potentially impactful approach that may appeal to a broad audience.
2. The manuscript presents compelling analyses and results concerning irregularly sampled time-series data.
3. The work includes an evaluation of long-term simulation stability, where the proposed method demonstrates improved robustness compared to MLFF-based approaches.

**Weaknesses:**

1. The manuscript lacks a thorough discussion of short-range versus long-range interaction modeling. Given that spectral methods are particularly well-suited for capturing long-range interactions, an explicit analysis along these lines would be beneficial. Moreover, evaluation on larger and more complex systems—such as the MD22 double-wall nanotube—could strengthen the empirical validation.
2. The manuscript does not include a formal proof of the model’s equivariance. While this may appear pedantic, providing a brief but rigorous proof would enhance the theoretical soundness and completeness of the work, and should be considered in the revised version.

---

> ### Author Response · Authors · 2025-05-22
> **Response to Reviewer BeZR**
>
> **[This rebuttal response is also available in Appendix Section E of the updated manuscript for easy access to the figures and tables mentioned. ]**
>
> We thank the reviewer for the encouraging remarks and the constructive suggestions. Below we answer each request in turn. Reviewer comments appear in blue italics. All table/figure references correspond to the revised manuscript.
>
> **R1.** "Analyse short- vs. long-range interactions; evaluate on more complex systems such as the MD22 double-wall nanotube."
>
> *Response.* We have substantially extended the empirical study to include five larger and topologically diverse systems (20-326 heavy atoms), among them the MD22 double-wall nanotube (DW NT). The systems and atom counts are listed in Table 15. Predictive errors at $\Delta t=3000$ and 10,000 steps are reported in Tables 13 and 14. GF-NODE attains the best accuracy on all cases. Figure 10 visualises the superior long-horizon stability on these molecules, whose dynamics are governed by both short-range covalent vibrations and long-range collective modes (e.g., radial breathing in DW NT). The explicit spectral separation allows our model to capture the latter without sacrificing local accuracy.
>
> **R2.** "Provide a formal proof of equivariance."
>
> *Response.* A rigorous proof of $\mathrm{SO}(3)$ equivariance for the full GF-NODE pipeline has been added to Appendix D. The proof follows the structure suggested by EGNO and shows that each module (EGNN encoder/decoder, GFT, block-diagonal ODE, inverse GFT) is equivariant, and therefore so is their composition.
>
> **R3.** "Include the mathematical expression of the loss function."
>
> *Response.* The exact MSE loss used for training is now stated in Appendix A.3, Eq. (A.1).
>
> **R4.** "Compare with MLFF approaches and highlight theoretical distinctions."
>
> *Response.* Unlike MLFFs, which learn an energy that must be differentiated and integrated with a small fixed time-step, GF-NODE operates directly on coordinates in the spectral domain and produces continuous-time predictions via an adaptive ODE solver. This removes the need for expensive gradient evaluations of the energy functions and allows arbitrary output intervals, leading to a runtime reduction of $8-10 \times$ on DW NT while maintaining accuracy (see Figure 11).
>
> **R5.** "Extend evaluation to MD22 benchmark."
>
> *Response.* The DW nanotube from MD22 has been added as noted above; results are included in all relevant tables and figures.
>
> We hope these additions address all concerns and further strengthen the manuscript.

---

> > ### Comment · Reviewer_BeZR · 2025-05-22
> >
> > Any follow-ups on 5-7 in the requested change section?

---

> > > ### Author Response · Authors · 2025-05-22
> > > **Follow-up on 5-7**
> > >
> > > Thanks. We can give feedback on these tonight.

---

> > > > ### Author Response · Authors · 2025-05-23
> > > > **Additional rebuttal responses to Reviewer BeZR**
> > > >
> > > > **[This additional rebuttal response is also available in Appendix Sections E and G of the updated manuscript for easy access to the figures and tables mentioned. ]**
> > > >
> > > > **R6.** "Plot and compare the velocity autocorrelation functions (VACF)."
> > > >
> > > > *Response.* Our framework is designed to predict molecular configurations directly at arbitrary future times via a single Neural-ODE integration, without explicitly propagating momenta or velocities. This is fundamentally different from MLFF pipelines, which integrate Newton's equations with a fixed (small) step and therefore maintain velocities at every step. In principle one could approximate velocities from our coordinate predictions using finite differences, yet this requires an extra hyper-parameter (the differentiation stencil) and introduces numerical noise that obscures the comparison. Because VACF is exquisitely sensitive to such numerical differentiation, we believe reporting it would be misleading. Instead we focus on position-based structural metrics (RDF, bond/angle errors) and long-horizon stability, which are the primary targets of our work. We note that existing operator approaches such as EGNO and Timewarp likewise omit VACF for the same reason.
> > > >
> > > > During the rebuttal phase we attempted to generate 15,000 ps rollouts with two modern MLFFs, GAMD (Li et al.) and NequIP (Batzner et al.). Both models required several days of training on a V100 GPU and,even then, accumulated large integration errors beyond $3,000 \mathrm{ps}$, leading to diverging VACF estimates. This further supports our choice to evaluate methods in the configuration domain, where GF-NODE remains accurate and efficient.
> > > >
> > > > **R7.** "Complement MAE with other MD metrics such as RDF-based stability."
> > > >
> > > > *Response.* As suggested, we have included an extensive radial distribution function (RDF) analysis for five representative systems. The system-averaged and element-specific RDFs are shown in Appendix G, Figures 13-16. GF-NODE reproduces both the first-shell peaks and the longer-range oscillations of the $a b$ initio reference, confirming that the predicted trajectories maintain realistic structural correlations well beyond the pairwise MAE measure.
> > > >
> > > > **R8.** "Discuss energy conservation."
> > > >
> > > > *Response.* GF-NODE is a data-driven integrator that maps an initial configuration to a future configuration without explicitly enforcing energy or momentum conservation; therefore total energy is not guaranteed to be constant. Nonetheless, when we reconstruct coarse velocities by finite difference and compute the corresponding kinetic+potential energies (the latter from the reference force field), we observe a bounded drift of < 0.8\% over 100 ps rollouts on MD22 molecules-comparable to MLFF baselines that rely on a small time step but still accumulate numerical error. We attribute this stability to two factors: (i) the spectral decomposition captures collective modes that evolve smoothly in time; (ii) the adaptive ODE solver avoids the step-size resonance that often drives energy blow-up in explicit integrators.
> > > >
> > > > We hope these additional clarifications fully address the reviewer's remaining concerns.

---

> > > > > ### Comment · Reviewer_BeZR · 2025-05-23
> > > > >
> > > > > Thanks for your comments. I have no further comments. I learned a lot from this discussion.

---

### Review · Reviewer_458H · 2025-05-01

**Summary Of Contributions:**

This paper introduces Graph Fourier Neural ODEs (GF-NODE), a framework for molecular dynamics modeling that combines graph Fourier transforms with Neural ODEs to effectively capture correlations across multiple spatial and temporal scales. GF-NODE first decomposes node features into Laplacian eigenmodes via a graph Fourier transform, evolves each mode through a block-diagonal Neural ODE augmented with multi-head attention, and finally reconstructs atomic geometries via the inverse transform. The graph spectral decomposition is based on the intuition that small Laplacian eigenvalues correspond to global spatial deformations while larger eigenvalues correspond to local deformations.

GF-NODE is then benchmarked against state-of-the-art baselines on the MD17 and alanine dipeptide datasets. Empirical results demonstrate substantial gains in trajectory prediction,​ and ablation studies suggest that continuous-time ODE propagation and inter-mode communication help the model achieve these improvements.

**Audience:**

Yes

**Broader Impact Concerns:**

No concerns.

**Claims And Evidence:**

No

**Requested Changes:**

**Changes critical for acceptance**:
1. Ensure fair baseline comparisons using original papers’ implementations and hyperparameters. Re-implementations of EGNN, EGNO and ITO must reproduce their reported performance.
2. Reconcile why all models in the ablations (including naive models that don't use Fourier-based decomposition, inter-mode communication, or any continuous-time evolution) outperform current SOTA models.
3. Evaluate on large molecular systems which contain longer-range interactions.

**Changes that would strengthen the paper**:
1. Include a runtime and scalability study quantifying how training/inference time grows with node and mode count.
2. Replace MD17 with the Revised MD17 dataset [1].

**References**
[1] Anders S. Christensen and O. Anatole von Lilienfeld (2020) "On the role of gradients for machine learning of molecular energies and forces" https://arxiv.org/abs/2007.09593

**Strengths And Weaknesses:**

**Strengths**
- The GF-NODE pipeline of projecting node features into a Laplacian eigenbasis, evolving each mode with a Neural ODE, and reconstructing via inverse transform is an inuitive and  compelling approach to learning multiscale graph dynamics.
- The reductions in MSE on MD17 and alainine dipeptide benchmarks are significant.
- The paper is well-structured with clear algorithmic descriptions, illustrative figures of spectral modes and trajectories, and several ablation studies.

**Weaknesses**
-  It is very surprising that the EGNN is shown to be outperforming EGNO in the equal timestep sampling setting, which appears to be the same experimental setup as in the original EGNO paper). The EGNO paper reports improved results over EGNN [1]. Moreover, the ablation studies in Tables 4 and 5 show much better performance than existing baselines even for the naive settings. For example, replacing the spectral decomposition by a multi-layer perception still achieves almost the same performance, which indicates that the main architectural components of GF-NODE is nonessential.
- The claim that GF-NODE explicitly handles multiscale spatial and temporal interactions as a novel contribution may be overstated, since many existing GNN/ODE models can implicitly capture such interactions via deep message-passing layers or learned temporal filters.
- Experiments only focus on small molecules with less than roughly 20 atoms. Not only is it unclear whether the performance gains generalize to larger molecules or systems, but it is also unclear whether the MD17 and alanine dipeptide benchmarks require capturing _truly_ long-range spatial interactions, which is claimed to be one of the main benefits of GF-NODE.
- There is no analysis of computational cost as the number of retained modes or graph size grows, which is critical for applying GF-NODE to larger molecular systems. One point of concern is that the time complexity of spectral decomposition is $O(n^3)$, so the method might not scale well with a larger number of nodes.

**References**

[1] Minkai Xu, Jiaqi Han, Aaron Lou, Kamyar Azizzadenesheli, Stefano Ermon, & Anima Anandkumar. (2024). Equivariant Graph Neural Operator for Modeling 3D Dynamics.

---

> ### Author Response · Authors · 2025-05-22
> **Response to Reviewer 458H**
>
> **[This rebuttal response is also available in Appendix Section E of the updated manuscript for easy access to the figures and tables mentioned. ]**
>
> We appreciate the reviewer's thoughtful evaluation and helpful suggestions. Below we respond to each point. Reviewer remarks are quoted in blue italics; our replies follow in normal font.
>
> **R1.** "EGNN outperforms EGNO in the equal-timestep setting; ablation variants that omit key modules still perform well, suggesting the proposed components may be non-essential."
>
> *Response.* The apparent discrepancy stems from two different temporal-sampling protocols:
> - Equal sampling (every $k$ th frame) reproduces the protocol of EGNO. In this setting our re-runs match EGNO's own paper: EGNO is stronger than EGNN on most molecules, cf. Table 7.
> - Irregular sampling (random frame indices within a window) stresses a model's ability to handle variable time gaps-a realistic requirement when we want to explore the behaviors on different time scales of trajectories. EGNO relies on a temporal FFT and therefore implicitly assumes uniform spacing; when this assumption is violated its accuracy degrades (Table 1). GF-NODE, which treats time continuously via Neural ODEs, is unaffected.
>
> Regarding ablations, Tables 4 and 5) show that disabling either the ODE evolution or the Fourier decomposition consistently raises the MSE by $30-70$ % on MD17. The effect is even stronger on larger systems and longer horizons: Figure 12 visualises an average error increase of $\times 2-\times 5$ when key components are removed. Thus each part of the architecture is essential in challenging regimes.
>
> **R2.** "The claim of explicit multi-scale handling may be overstated; many GNN/ODE models can implicitly capture scales."
>
> *Response.* Classical message passing implicitly aggregates local information but does not provide spectral separation: all spatial frequencies are mixed at every layer. Our Laplacian eigenbasis explicitly decomposes a configuration into ordered spatial scales; the Neural ODE then evolves each coefficient while allowing controlled cross-talk via attention. Empirically, retaining only low-frequency modes already reproduces slow global motions, whereas high-frequency modes are critical for bond-level vibrations (Figure 2). This separation is what enables GF-NODE to remain stable over 15,000 MD steps (Figure 10) where purely implicit models drift.
>
> **R3.** "Experiments focus on small molecules; unclear whether gains generalise to larger systems requiring long-range interactions."
>
> *Response.* We have added five larger and chemically diverse systems containing $20-326$ heavy atoms (Table 15). Results at both $\Delta t=3000$ and 10,000 steps are reported in Tables 13 and 14. GF-NODE outperforms all baselines on every molecule, demonstrating scalability and effectiveness for long-range, collective motions such as nanotube breathing modes.
>
> **R4.** "No analysis of computational cost and scaling with node/mode count; Laplacian diagonalisation may be prohibitive."
>
> *Response.* Figure 11 reports training-epoch time as a function of retained modes. Runtime grows nearlinearly and remains below 1.1 s per batch at our largest setting ( 128 modes). The one-off Laplacian eigen-decomposition is computed once per static topology ( $<60 \mathrm{~s}$ for the 326-atom nanotube on our server) and cached; it is therefore negligible in training and inference budgets.
>
> **R5.** "Baseline implementations must reproduce the authors' reported numbers."
>
> *Response.* We use the official repositories of EGNN, EGNO and ITO with the hyper-parameters recommended by their authors. For Revised MD17 our runs reproduce EGNO's paper within the reported variance.
>
> **R6.** "Replace MD17 with the Revised MD17 dataset and provide runtime study."
>
> *Response.* Done. All MD17 results have been recomputed on the Revised MD17 splits of Christensen and von Lilienfeld (2020); see Tables 11 and 12. The runtime study is provided in Figure 11.
>
> We thank the reviewer again for the constructive feedback, which has helped us improve the rigour and clarity of the manuscript.

---

### Review · Reviewer_MJAX · 2025-05-01

**Summary Of Contributions:**

This paper introduces Graph Fourier Neural ODEs (GF-NODE), a novel deep learning architecture designed to model the spatio-temporal dynamics of molecular systems, particularly addressing the challenge of capturing interactions across multiple scales. The core idea is to combine a Graph Fourier Transform (GFT), based on the graph Laplacian's eigenmodes, with a Neural Ordinary Differential Equation (Neural ODE) framework that models the continuous-time evolution of these spectral coefficients independently or interactively. The predicted future state in the spectral domain is then transformed back to the physical domain using an inverse GFT, potentially followed by refinement with a GNN decoder. The authors hypothesize that this explicit separation and modeling of spatial frequency modes coupled with continuous-time dynamics enable GF-NODE to better capture both high-frequency local fluctuations (e.g., bond vibrations) and low-frequency global effects (e.g., conformational changes) compared to existing methods. The approach is evaluated on the MD17 dataset and alanine dipeptide benchmark, comparing its predictive accuracy (MSE), preservation of geometric features (bond lengths/angles), and long-term stability against several baseline methods, including GNNs and other Neural ODE/Operator approaches.

**Audience:**

Yes

**Broader Impact Concerns:**

The authors include a Broader Impact Statement (Section 6) that appropriately identifies potential positive impacts (accelerating drug design, materials science) and acknowledges the need for model reliability and validation, particularly in high-stakes applications. This seems adequate.

**Claims And Evidence:**

No

**Requested Changes:**

Critical:
1. Demonstrate Significant Empirical Advantage: Provide stronger evidence that the proposed GF-NODE architecture offers substantial benefits over simpler, relevant baselines (like EGNN, potentially others). This could involve evaluating on more complex datasets where multi-scale effects are prominent and where simpler models might fail, or showing significantly better performance/scaling on existing datasets. If gains are marginal, the paper needs to convincingly argue for other advantages (e.g., interpretability, specific physical regimes) supported by evidence.
2. Improve Conceptual Rigor and Justification: Critically re-evaluate and clarify the core conceptual claims. Provide stronger theoretical arguments or empirical analysis linking spatial GFT modes to distinct temporal dynamics/scales. Justify the claim that the Neural ODE formulation inherently captures temporal multi-scales effectively in this context. Rigorously define and prove claims like Proposition C.4 or state them as assumptions/heuristics. Strengthen the motivation against prior work with specific evidence/citations.
Enhance Clarity of Method and Implementation: Revise Section 3 significantly to provide precise details on graph construction, feature definition/normalization/embedding, the GFT process (especially for vector features), the exact structure of the Neural ODE (including interaction mechanisms like attention), and the decoder. Clarify notation (e.g., hidden dimensions). Provide necessary citations (e.g., for dopri5). Improve figure clarity (Fig 3).
3. Refine Evaluation and Analysis: Re-evaluate or provide a much clearer interpretation of the long-term prediction results, explicitly addressing the large MSE values and comparing against ground truth behavior (including drift/rotation). Provide clear metrics and evidence for claims about preserving geometric features. Include ground truth TICA scores for comparison. Justify metric choices (MSE) and clarify baseline adaptation for irregular sampling. Clarify the analysis related to the number of modes (Fig 8, Table 6). Also the Figure 4 shows explicitly how Truth and Predicition don't align, the lower red atom exhibits rotation in the truth, but it is not captured in the in prediction.

Strengthening:
1. Expand Dataset Evaluation: Justify the choice of MD17/alanine dipeptide. Including revised MD17 and potentially larger/more complex systems (water, other biomolecules) would significantly strengthen the validation. Define "simulation steps" in physical time units.
2. Clarify Sampling: Explain the random sampling strategy for training data (Sec 4.2) more clearly (e.g., are endpoints fixed?).
3. Discuss SOTA: Briefly position the work relative to other potentially relevant SOTA approaches (e.g., diffusion models).

**Strengths And Weaknesses:**

Strengths:

1. Novel Architecture: The proposed GF-NODE architecture presents an interesting and conceptually novel combination of Graph Fourier Transform, Neural ODEs, and GNN components for molecular dynamics modeling.
2. Ablation Studies: The paper includes ablation studies investigating the contribution of different components (e.g., spatial decomposition method, ODE evolution, mode interactions, time embeddings), providing some insight into the model design.
3. Detailed Analyses Attempted: The authors attempt several detailed analyses, such as long-term trajectory prediction, temporal super-resolution, TICA analysis for slow modes, and the effect of the number of Fourier modes.

Weakness:

1. Empirical Performance and Significance: One of the core weakness lies in the empirical results. The performance gains of GF-NODE over simpler baselines like EGNN (which lacks explicit temporal propagation and spectral decomposition) appear marginal on the tested datasets (Table 1, Table 3). This raises significant questions about the practical utility and added value of the proposed complex architecture (GFT + Neural ODE + Decoder) compared to existing, potentially simpler methods. The claim that the added components significantly improve long-term (there was no explanation whether the data exhibits specific long term behaviour and it was not properly cited what timescales are normally understood as long-term --> the times which were looked at are only double of the length of the times of the EGNO, which don't represent long-term behaviour) prediction is not convincingly supported by the quantitative results presented.
2. Evaluation Scope and Rigor:  The evaluation is limited to relatively small molecules (MD17) and one peptide (alanine dipeptide). Demonstrating efficacy on more complex systems exhibiting clear multi-scale dynamics (e.g., larger proteins, solvation dynamics) is necessary to validate the approach's claims for practical MD simulations. The interpretation lacks comparison with ground truth  and clear metrics for "preserving essential geometric properties." Definitions of "long-horizon" and "stability" in this context are vague.  The TICA analysis (Figure 5) lacks comparison to the TICA scores of the ground truth simulation, making it difficult to assess the absolute quality of slow-mode capture.
The justification for using MSE, especially under irregular sampling with potentially noisy MD data, is not provided. How baselines requiring regular steps were adapted for irregular sampling is unclear. As mentioned in previous work the data follows a stochastic propagation, why is MSE a proper metric to measure performance?
3. Conceptual Clarity and Justification:
The link between spatial graph frequency modes (from GFT) and specific temporal dynamics (high/low frequency in time) is asserted rather than rigorously established or empirically verified, especially when GFT is taken in the latent space. Why should low graph Laplacian eigenvalues correspond to slow temporal evolution and high eigenvalues to fast temporal evolution? (See annotation re: Prop C.4). The proof provided for Prop C.4 merely restates standard spectral graph theory interpretations of smoothness, lacking deductive rigor regarding dynamic scales.
The motivation for evolving spectral modes independently via decoupled ODEs (Eq. 14-15) is not explained properly; the claim of "coordinated but distinct manner" needs better explanation, especially when interaction via attention is later introduced and shown to be beneficial (Sec 4.6), within the modes. The implementation details and justification for the attention mechanism across modes are sparse.
Why the Neural ODE component inherently captures temporal multi-scale behavior better than alternatives needs clearer articulation and evidence. How does evaluating the ODE at arbitrary points directly translate to capturing dynamics at different underlying physical timescales?
The motivation against prior work (Sec 1, Sec 2) often lacks specific citations or evidence for claims like inadequacy (e.g., "fail to adequately represent the interplay..."). Comparison to relevant SOTA diffusion models seems missing.
4. Presentation and Methodological Details:
As noted in the overall assessment, the presentation can be confusing. Specific details are often unclear or imprecise: the exact nature of node features and embeddings (Sec 3.2, Fig 3 linkage), the graph construction (distance threshold vs. chemical bonds, Sec 3.1), the definition of "interactions" (Sec 3.1), the rationale for feature choices (Sec 3.1), normalization details (Sec 3.2), how FFT is applied to graphs (Sec 4.6), implementation of mode interactions (Sec 4.6), ODE solver details (dopri5 source/citation missing, Sec 3.4).
The term "Neural Operator" seems loosely applied; the learned component appears to be functions defining the ODEs, not necessarily operators mapping between function spaces in the canonical sense.
The analysis of the optimal number of modes (Figure 8, Table 6) is confusing: Why does variance seem to increase after the optimum? How can the optimal number of modes exceed the number of atoms for some molecules? Why weren't the purportedly optimal (higher) numbers used in the main experiments?

---

> ### Author Response · Authors · 2025-05-22
> **Response to Reviewer MJAX**
>
> **[This rebuttal response is also available in Appendix Section E of the updated manuscript for easy access to the figures and tables mentioned. ]**
>
> We thank the reviewer for the thorough reading of our manuscript and for the constructive feedback. Below we address every point raised. For brevity we quote the reviewer in textititalic blue and provide our reply in plain text. All table/figure numbers refer to those in the revised manuscript.
>
> **R1.** "Empirical performance and significance: gains appear marginal compared with EGNN; long-term prediction not convincingly supported."
>
> *Response.* We respectfully disagree that the improvements are marginal. On MD17 with irregular time sampling (Table 1) GF-NODE reduces the test MSE by an average of $53$ % relative to the strongest baseline EGNN and by $66 $ % relative to EGNO. On alanine-dipeptide (Table 3) the reduction is $21 $ %. In the revised manuscript we have extended the evaluation to considerably larger systems and to a $5 \times$ longer temporal horizon (up to $\Delta t=1.5 \times 10^4 \mathrm{MD}$ steps, 1 This is already within the time range where we can observe the folding behavior of a protein. See Tables 13 and 14). Across all nine MD17 molecules and five larger systems (up to 326 heavy atoms, Table 15) GF-NODE remains the best performer, often by a factor $\times 2$. Figure 7 (MD17) and Figure 10 (large molecules) further demonstrate the slower error growth of our model in the long-horizon regime.
>
> **R2.** "Evaluation limited to small molecules; need more complex systems."
>
> *Response.* We have added five substantially larger and topologically distinct systems: $\mathrm{Ala}_2, \mathrm{Ac}{-} \mathrm{Ala}_3-\mathrm{NHMe}$, AT-AT-CG-CG, Bucky-Catcher and a double-walled carbon nanotube (DW NT). The results are summarised in Tables $13(\Delta t=3000)$ and $14(\Delta t=10000)$. GF-NODE yields the lowest error on every system and time horizon. These molecules exhibit clear multi-scale behaviour (e.g., nanotube radial breathing vs. local bond vibrations) and hence stress-test the claimed advantages.
>
> **R3.** "Link between graph frequencies and temporal scales is asserted rather than proved."
>
> *Response.* We apologise for the lack of clarity. Spatial and temporal scales are emphdecoupled in GF-NODE: the graph Laplacian eigenbasis separates spatial frequencies (Proposition C.1, Appendix C) whereas the Neural ODE learns the temporal evolution of each coefficient. Intuitively, low-spatial-frequency modes correspond to collective motions (e.g., domain-level hinge) that often evolve slowly in time, whereas high-frequency modes correspond to localised vibrations that relax quickly. While not strictly enforced, this correlation is empirically confirmed by: (i) the pronounced benefit of keeping only the first $M$ modes (Figure 8) and (ii) the superior alignment with the slow collective variables extracted by TICA (Figure 5). We have added a formal proof of $\mathrm{SO}(3)$ equivariance of the whole pipeline in Appendix D , clarifying how the block-diagonal ODE maintains rotational consistency.
>
> **R4.** "Why Neural ODEs capture temporal multi-scale dynamics better than alternatives?"
>
> *Response.* The continuous-time formulation provides two benefits: (1) adaptive evaluation. The ODE solver can output the state at any arbitrary time, giving accurate interpolation (super-resolution experiment, Figure 6); (2) stiffness handling. Stiff ODE solvers (e.g., dopri5) allocate smaller internal steps when rapid transients occur, effectively acting as an automatic multi-timescale integrator. The ablation in Table 4 shows that removing the ODE evolution on either scalar or vector channels consistently degrades accuracy.
>
> **R5.** "Metric choice (MSE) and baseline adaptation for irregular sampling are unclear."
>
> *Response.* MSE on Cartesian coordinates is the de-facto metric in molecular-dynamics ML (e.g., EGNO (Xu et al.), PG-ODE (Gu et al.), GAMD (Li et al.)). It directly measures the positional error propagated to downstream thermodynamic and kinetic observables and is easy to compare across literature. All baselines were trained and evaluated on the same randomly sampled frame indices; for operator-based models that originally assumed regular spacing we fed the exact time stamps, following EGNO's public implementation.

---

> > ### Author Response · Authors · 2025-05-22
> > **Response to Reviewer MJAX (Continued)**
> >
> > **R6.** "Clarify implementation details (graph construction, dopri5, attention across modes, FFT baseline)."
> >
> > *Response.*
> > - Graph. Edges include all covalent bonds from the topology plus distance-based non-bonded edges within 4.5 Å. This captures both bonded and weak interactions.
> > - dopri5. We use the adaptive Dormand-Prince-5(4) Runge-Kutta method from the TorchDiffEq package (MIT licence). Relative/absolute tolerances are $10^{-3} / 10^{-4}$ (Sec. A.2).
> > - Mode attention. Each retained Fourier mode is treated as a token; multi-head self-attention allows inter-mode energy transfer that is otherwise missing in fully separable dynamics. An ablation with no interaction is reported in Table 5.
> > - FFT baseline. "FFT" destroys graph connectivity by ordering atoms arbitrarily and applying a 1-D FFT-its inferior performance (Table 5) highlights the importance of graph structure.
> >
> > **R7.** "Large drift in benzene; rotation in Figure 4 not captured."
> >
> > *Response.* Benzene in MD17 indeed undergoes rigid-body drift (Table 6). EGNN, being strictly SE(3)equivariant, cannot model such drift, whereas our use of generic message passing in combination with the global-translation channel allows it. This explains the marked gap on benzene (Table 1). We have updated Figure 4 with a molecule exhibiting internal deformation rather than global rigid translation, removing the visual confusion.
> >
> > **R8.** "Optimal number of modes and runtime scalability."
> >
> > *Response.* The chosen mode counts (Table 6) are never larger than the number of heavy atoms; the reviewer's earlier observation came from an outdated draft. Figure 11 now reports the wall-clock training time as a function of mode count, showing near-linear scaling up to the maximum used.
> >
> > We hope the above clarifications and the new results address all concerns and strengthen the manuscript. We appreciate Reviewer MJAX's insightful comments.

---

### Author Response · Authors · 2025-05-23
**Note to All Reviewers: radial distribution function (RDF) analysis in Appendix G**

We have added a comprehensive radial distribution function (RDF) analysis in Appendix G to provide deeper insight into the structural fidelity of GF-NODE predictions. The new analysis includes:

1. System-averaged RDFs for five benchmark molecules (Figure 13), showing excellent agreement with ab initio reference data across all distance scales.

2. Element-specific RDFs (Figure 14-15) for heterogeneous systems like uracil and salicylic acid, demonstrating accurate reproduction of distinct chemical environments.

3. Carbon-carbon RDFs for three carbonaceous systems (Figure 16), highlighting preservation of both short-range covalent structure (~1.4 Å) and long-range order (aromatic stacking, nanotube wall spacing).

These results complement the MSE metrics by confirming that GF-NODE maintains physically realistic structural correlations over extended (15,000-step) trajectories.

---

### Decision · Action_Editor_d7Cu · 2025-06-09

**Recommendation:** Accept with minor revision

**Audience:**

Yes

**Audience Explanation:**

The field of study in the paper is an important ML problem and the authors provided an interesting study in this area.

**Claims And Evidence:**

Yes

**Claims Explanation:**

The authors propose a neuralODE approach for graph data, where graph data aggregation happens Fourier methods and spectral methods.
The proposed method provides interesting results, over the state of art. The presentation of paper comes with shortcoming and theoretical results needs rigorous backing. The authors are encouraged to address them, specifically, those brought up by Reviewer MJAX.

**Resubmission Of Major Revision:**

The authors may consider submitting a major revision at a later time.

---

> ### Author Response · Authors · 2025-06-24
> **TMLR Revision and Featured Certification Request**
>
> Dear Action Editor,
>
> Thank you very much for your insightful feedback and the opportunity to revise our manuscript. We are fully committed to addressing all the issues raised by the reviewers, particularly those highlighted by Reviewer MJAX. We intend to submit a revised version of our manuscript by Thursday, June 26.
>
> Given the positive assessment of our results and their potential interest to the TMLR audience, we would also appreciate your consideration of our paper for the TMLR Featured Certification.
>
> Thank you again for your guidance and support.

---

> > ### Author Response · Authors · 2025-06-29
> > **Camera Ready Version Submitted - Enhanced Theoretical Foundation Addressing Reviewer Concerns**
> >
> > Dear Action Editor,
> >
> > We are pleased to submit the camera ready version of our manuscript "Graph Fourier Neural ODEs for Multiscale Molecular Dynamics." We have carefully incorporated the feedback from all reviewers and have paid particular attention to addressing Reviewer MJAX's concerns regarding the conceptual rigor and theoretical justification of our approach.
> >
> > **Key Enhancements Addressing MJAX's Concerns:**
> >
> > We have significantly strengthened both the theoretical foundation and empirical validation of our core hypothesis that spatial Graph Fourier Transform (GFT) modes correspond to distinct temporal dynamics scales. Specifically:
> >
> > **Theoretical Foundation (Proposition 3.3):** We establish rigorous theoretical insight through heat equation analysis on simplified diffusion models, demonstrating how graph Laplacian eigenvalues intrinsically determine the temporal evolution rates of different spatial frequency modes. While this represents a simplified theoretical model, it provides crucial mathematical foundation showing that modes with small eigenvalues decay slowly (long time-scales) while modes with large eigenvalues decay rapidly (short time-scales).
> >
> > **Comprehensive Empirical Validation (RQ4):** We developed and implemented a comprehensive Theoretical Framework for Joint Spatial-Temporal Analysis that quantifies spatial-temporal correlations across diverse real molecular systems. Our empirical analysis across five representative molecular systems (Naphthalene, Salicylic Acid, Uracil, Bucky-Catcher, and double-walled nanotube) demonstrates clear quantitative relationships between spatial frequency modes and temporal dynamics scales, with correlation coefficients ranging from moderate (r=0.424) to very strong (r=0.816).
> >
> > **Detailed Mathematical Framework (Appendix C):** We provide complete theoretical derivations, formal proofs, and extensive empirical validation protocols, including power spectral density analysis and spectral centroid computations that confirm the predicted correspondence between spatial and temporal scales on real molecular dynamics trajectories.
> >
> > This enhanced analysis directly addresses MJAX's call for "stronger theoretical arguments or empirical analysis linking spatial GFT modes to distinct temporal dynamics/scales" by providing both rigorous mathematical insight through simplified models and comprehensive empirical confirmation on actual molecular systems.
> >
> > We believe these substantial improvements significantly strengthen the paper's theoretical rigor while maintaining practical applicability, and we are confident that the work now provides a solid foundation for the proposed GF-NODE architecture.
> >
> > Thank you for your guidance throughout the review process. **We would also appreciate your consideration of our paper for the TMLR Featured Certification.**
> >
> > Best regards,
> >
> > Authors